# Effect of SARS-CoV-2 prior infection and mRNA vaccination on contagiousness and susceptibility to infection

Denis Mongin [1], Nils Bürgisser [1,2], Gustavo Laurie[3], Guillaume Schimmel[3], Diem-Lan Vu[1,3,4,5], Stephane Cullati [6,7], Covid-SMC Study Group* & Delphine Sophie Courvoisier[1,6]

The immunity conferred by SARS-CoV-2 vaccines and infections reduces the transmission of the virus. To answer how the effect of immunity is shared between a reduction of infectiousness and an increased protection against infection, we examined >50,000 positive cases and >110,000 contacts from Geneva, Switzerland (June 2020 to March 2022). We assessed the association between secondary attack rate (i.e. proportion of new cases among contacts) and immunity from natural infection and/or vaccination, stratifying per four SARS-CoV-2 variants and adjusting for index cases and contacts' socio-demographic characteristics and the propensity of the contacts to be tested. Here we show that immunity protected contacts from infection, rather than reducing infectiousness of index cases. Natural infection conferred the strongest immunity. Hybrid immunity did not surpass recent infection. Although of smaller amplitude, the reduction in infectiousness due to vaccination was less affected by time and by the emergence of new SARS-CoV-2 variants than the susceptibility to infection. These findings support the role of vaccine in reducing infectiousness and underscore the complementary role of interventions reducing SARS-CoV-2 propagation, such as mask use or indoor ventilation.

Since its worldwide spread at the beginning of 2020[1], the SARS-CoV-2 virus has caused one of the most important health burdens in recent history. It is estimated to have caused 18 million deaths as of end of 2021[2]. SARS-CoV-2 became a leading cause of death in some countries in these years[3] and is responsible for an important burden of long-lasting symptoms in the population[4]. Its widespread circulation within human communities and possible animal reservoirs[5] allows the SARS-CoV-2 to mutate frequently[6] and has resulted so far in more contagious, immunity-escaping variants[7,8] responsible for successive waves of infections worldwide.

The effect of immunity on the transmission of the successive SARS-CoV-2 variants and its evolution in time are key factors for our understanding of the SARS-CoV-2 propagation. Immunity can be acquired through vaccination or through natural infection. SARS-CoV-2 mRNA vaccines have been shown to be effective in preventing re-infection shortly after administration[9]. However, the immunity they

[1]Faculty of Medicine, University of Geneva, Geneva, Switzerland. [2]General internal medicine division, Department of Medicine, Geneva University Hospitals, Geneva, Switzerland. [3]Division of General cantonal physician, Geneva Directorate of Health, Geneva, Switzerland. [4]Division of Infectious Diseases, Geneva University Hospitals, Geneva, Switzerland. [5]Laboratory of Virology, Division of Laboratory Medicine, Geneva University Hospitals, Geneva, Switzerland. [6]Division Quality of care, University Hospitals of Geneva, Geneva, Switzerland. [7]Population Health Laboratory (#PopHealthLab), Faculty of Science and Medicine, University of Fribourg, Fribourg, Switzerland. *A list of authors and their affiliations appears at the end of the paper. ✉e-mail: denis.mongin@unige.ch

confer wanes rapidly[10–12] and a roll-out of booster vaccinations has been implemented in high-income countries to maintain an immunity against SARS-CoV-2[13–15]. It was recently demonstrated that natural infection confers a stronger and longer-lasting protection against reinfection than vaccination[7,16–18], and that the combination of both type of immunity (hybrid immunity) may provide an even stronger protection[18,19]. Less is known, however, on the effect of immunity on the probability to contaminate others (infectiousness), especially with regard to natural immunity[19]. Depending on the variant of concern (VoC) considered, studies analysing the secondary attack rate show contrasting results, from no effect of vaccination on infectiousness[20–22] to a clear reduction in the attack rate[23]. The effect of previous infection on the reduction of infectiousness and its evolution over time is unclear, while recent in vitro studies measuring viral load and propagation indirectly suggest that natural infection could reduce infectiousness better than vaccination[24,25]. Similarly, little is known about how the reduction of infectiousness and the reduction of susceptibility to infections conferred by the immunity compare in the reduction of SARS-CoV-2 transmission. Secondary attack rate (SAR) is a good measure of SARS-CoV-2 transmission, providing a full picture of both the reduction of susceptibility and infectiousness that the immunity may confer. Apart from the immunity of the population and the VoC considered, SAR is known to vary greatly by contact settings, ranging from 20% in households to 6% in social gatherings during the first year of the pandemic[26–29], but also by the symptoms of the index cases[30,31] and the socio-demographic characteristics of the studied population[29,31–34]. By definition, SAR also depends directly on the capacity to detect SARS-CoV-2 infections among contacts, including the propensity of the contacts to get tested.

Using a register dataset of 50,973 index cases having declared 111,674 contacts in the State of Geneva[35], we propose to study the effect of the immune status on SARS-CoV-2 secondary attack rate (SAR) across 4 SARS-CoV-2 variants, considering vaccination and natural infection of index and contacts while adjusting for demographic, social and health factors as well as the contact settings and the tendency to test for SARS-CoV-2.

## Results

During the period of interest (01-06-2020 to 01-03-2022), 65,161 infections were recorded among persons living in Geneva and who declared at least one contact person. Among them, 9,890 refused to share their data for research. 15,327 declared contacts also refused to share their data, removing an additional 4,298 infections. The resulting dataset consisted of 50,973 index cases and 111,674 declared contacts. The mean number of declared contact per infected person was 2.2 overall, with a net decrease during the Omicron period (1.6 mean contacts per index, see Table 1).

Index cases were at 73% adults between 18 and 64 years, 22% children and 4.6% adults older than 65 years. The proportion of children for the index cases tripled between the EU1 wave (11%) and the Delta wave (38%). Overall, children were overrepresented and adults >65 years underrepresented in our cohort when compared to the demographics of the Geneva state (18.5% of children and 16.5% of adults above 65 years in 2022 in Geneva, see supplementary material).

The index cases were contacted in average 1.4 days after their last encounter with their contacts, this delay increasing in time from 0.8 during EU wave up to 2 days during Omicron. The vast majority of the index cases had symptoms (94%), among whom more than half had cough (58%). The majority of the contacts reported by the index were persons sharing their home (63%), this percentage increasing up to 77% during the Omicron wave. Concerning the immunity status, the proportion of vaccinated index cases increased from around 2% during the alpha wave, up to 52% during the Omicron wave, of which 25% had their last dose more than 6 months before the infection. Contacts were less

vaccinated (37% during omicron) and a higher proportion of them were previously infected (10%, compared to 2.9% for the index cases).

### Secondary attack rate

Among the 111,674 declared contacts, 46,417 performed a test during the 10 days following the date of the last contact with the index case and 21,435 had a positive test result (raw SAR of 19.2%). The number of tests performed by the contacts increased strongly during a period starting one day before the last contact with the index and decreasing back 10 days after (supplementary figure S1). There was no age difference between those who performed a test and those who did not (see Supplementary Table S1). The SAR depends on the delay during which we consider that a positive test of the contact indicates being infected by the index case. The raw SAR increased almost linearly of 3 percent point per day when increasing this delay from 0 to 8 days, to then plateauing after 10 days (see supplementary figure S2). For the rest of the study, a delay of 10 days was considered. The raw SAR changed across variant and was 16.4% during the EU1 wave, 20.9% during the alpha wave, 16.7% during the delta wave, and 26.3% during the omicron wave. Of note that the proportion of contact performing a test during the period of interest evolved in time: 31.7% during EU1, 63.1% during alpha, 40.5% during the delta wave and 42.1% during the omicron wave.

The reference category of the contact-index dyad for the adjusted model was defined as follow: two asymptomatic adult men, neither vaccinated nor with an antecedent infection (NVNI), between the age of 18 and 65, having a house contact in a building in a wealthy neighbourhood, of which the index case is not obese and not a vulnerable person, the contact person having performed one test in the past three months. For this reference category, the multivariable analysis yielded a SAR of 34.4% (95%CI: [31.8, 37.0]) for the EU1 variant, 29.9% (95%CI: [27.1, 32.7]) for the alpha variant, 32.6% (95%CI: [30.1, 35.1]) for the delta variant and 40.6% (95%CI: [36.9, 44.3]) for the omicron variant.

The main variables influencing the SAR (see Fig. 1 and Table 2) were the immune status of both the index case and the contact, the presence of symptoms or the presence of cough for the index case, the type of relation between the index case and their contacts, the age of the contacts, and the number of tests the contact had in the 3 months before the contact date. The age of the index case, as well as the index housing type had a limited effect on the SAR. The gender of both index and contacts, the obesity of the index, the index vulnerability or its neighbourhood socio-economic condition did not affect the SAR.

### Immune status

The SAR was decreased by an antecedent infection of the index case, with no obvious difference if the infection was recent, older than 6 months or hybrid (see supplementary table S2 and supplementary Figure S3). The reduction of SAR induced by an infection of the index case was of 10.5 adjusted percent points (pp) (95%CI: [7.0, 14.0]) during the EU1 variant wave, 8.6pp during the alpha wave [4.3, 12.8], 11.3pp [8.6, 14.0] during the delta wave and of 4.3pp [1.3, 7.3] during the Omicron wave (see Fig. 1 and Table 2). The effect of previous infection was stronger for the contacts, with a greater effect when the date of infection was less than 6 months before the index-contact date. Previous infection of less than 6 month or more than 6 months, respectively lowered the SAR of 12.6pp [10.5, 14.7] and 17.2pp [15.5, 19.0] for EU1, 26.3 pp [24.7, 27.8] and 19.7pp [17.1, 22.3] for alpha, 30.7pp [29.4, 32.1] and 15.2pp [13.3, 17.1] during delta and 31.9pp [30.0, 33.8] and 4.6pp [1.4, 7.7] during omicron. Considering an interaction between immune status and testing tendency, the decrease of infection susceptibility was around 6pp stronger for all variants if the contacts were tested at least once during the 3 months preceding their encounter with the index (see supplementary table S3 and supplementary figure S4). A vaccinated index was associated with a lower SAR across VoCs mainly when the last dose of vaccination was less than 6 months

**Table 1 | Socio-demographics characteristics of the index cases and declared contacts for the whole study period (Overall) and stratified per periods of variant predominance**

| | Overall | EU1 | Alpha | Delta | Omicron | Missing (%) |
|---|---|---|---|---|---|---|
| Number of index cases | 50973 | 17460 | 8140 | 11390 | 13983 | |
| Mean number of contact (SD) per index case | 2.19 (1.90) | 2.42 (2.33) | 2.49 (1.86) | 2.39 (1.88) | 1.56 (0.94) | |
| Delay [days] between test result and first contact (phone or form) (SD) | 1.39 (1.38) | 1.68 (1.72) | 0.77 (0.72) | 1.35 (1.09) | 2.05 (1.49) | 39.1 |
| Index who died from COVID-19 (%) | 206 (0.4) | 122 (0.7) | 49 (0.6) | 27 (0.2) | 8 (0.1) | 0 |
| Total number of contacts | 111674 | 42295 | 20311 | 27260 | 21808 | |
| Number of infected contacts within 10 days following contact (%) | 21435 (19.2) | 6922 (16.4) | 4238 (20.9) | 4545 (16.7) | 5730 (26.3) | 0.0 |
| Index Socio-economic- neighbourhood (%) | | | | | | 3.3 |
| Healthy | 39640 (36.3) | 14713 (35.4) | 7059 (35.1) | 10591 (40.1) | 7277 (34.7) | |
| Slightly vulnerable | 18154 (16.6) | 6570 (15.8) | 3585 (17.8) | 4249 (16.1) | 3750 (17.9) | |
| Moderatly vulnerable | 19533 (17.9) | 7575 (18.2) | 3717 (18.5) | 4446 (16.8) | 3795 (18.1) | |
| Highly vulnerable | 31728 (29.1) | 12713 (30.6) | 5729 (28.5) | 7114 (26.9) | 6172 (29.4) | |
| Index vulnerable person (%) | 8605 (7.9) | 3320 (8.0) | 2098 (10.4) | 2122 (8.0) | 1065 (5.1) | 2.2 |
| Index with symptoms (%) | 98254 (94.1) | 37671 (95.8) | 18464 (91.7) | 24273 (92.5) | 17846 (95.4) | 6.7 |
| Index with cough (%) | 60244 (57.7) | 21700 (55.2) | 11932 (59.2) | 14969 (57.1) | 11643 (62.2) | 6.7 |
| Index obesity (%) | 9351 (10.5) | 3544 (11.7) | 2168 (11.3) | 2045 (8.4) | 1594 (10.3) | 20.1 |
| Index women (%) | 59738 (53.6) | 22520 (53.3) | 10938 (53.9) | 14325 (52.8) | 11955 (55.1) | 0.2 |
| Index age category (%) | | | | | | 0.0 |
| 18–65 | 82118 (73.5) | 35576 (84.1) | 14769 (72.7) | 15891 (58.3) | 15882 (72.9) | |
| 0–17 | 24384 (21.8) | 4264 (10.1) | 4588 (22.6) | 10198 (37.4) | 5334 (24.5) | |
| 65+ | 5153 (4.6) | 2454 (5.8) | 954 (4.7) | 1165 (4.3) | 580 (2.7) | |
| Index immune status (%) | | | | | | 0.0 |
| Infected <6 months | 638 (0.6) | 252 (0.6) | 169 (0.8) | 46 (0.2) | 171 (0.8) | |
| Infected > 6 months | 1286 (1.2) | 66 (0.2) | 56 (0.3) | 351 (1.3) | 813 (3.7) | |
| hybrid | 1269 (1.1) | 0 (0.0) | 10 (0.0) | 139 (0.5) | 1120 (5.1) | |
| Non-vaccinated non-infected (NVNI) | 90156 (80.7) | 41977 (99.2) | 19721 (97.1) | 18995 (69.7) | 9463 (43.4) | |
| Vaccinated < 6 months | 11853 (10.6) | 0 (0.0) | 355 (1.7) | 5974 (21.9) | 5524 (25.3) | |
| Vaccinated > 6 months | 6471 (5.8) | 0 (0.0) | 0 (0.0) | 1754 (6.4) | 4717 (21.6) | |
| Index housing type (%) | | | | | | 2.2 |
| Building | 88690 (80.6) | 33623 (80.2) | 16333 (80.8) | 21113 (79.0) | 17621 (83.0) | |
| Single house | 18309 (16.6) | 6827 (16.3) | 3402 (16.8) | 5031 (18.8) | 3049 (14.4) | |
| Collective structure | 3105 (2.8) | 1492 (3.6) | 481 (2.4) | 584 (2.2) | 548 (2.6) | |
| Contact type (%) | | | | | | 38.2 |
| Same roof | 43993 (62.6) | 14402 (55.7) | 12121 (64.7) | 13315 (65.4) | 4155 (77.2) | |
| Intimate or familial | 19409 (27.6) | 8003 (31.0) | 5053 (27.0) | 5448 (26.8) | 905 (16.8) | |
| During the day | 6920 (9.8) | 3450 (13.3) | 1546 (8.3) | 1603 (7.9) | 321 (6.0) | |
| Contact female (%) | 56369 (51.7) | 21018 (51.0) | 10415 (52.0) | 13955 (52.7) | 10981 (51.4) | 2.3 |
| Contact immune status (%) | | | | | | 9.5 |
| Infected < 6 months | 5253 (5.2) | 1715 (4.4) | 1003 (5.3) | 1390 (6.2) | 1145 (5.5) | |
| Infected > 6 months | 1727 (1.7) | 177 (0.5) | 202 (1.1) | 585 (2.6) | 763 (3.6) | |
| hybrid | 3114 (3.1) | 0 (0.0) | 51 (0.3) | 1197 (5.3) | 1866 (8.9) | |
| Non-vaccinated non-infected (NVNI) | 77005 (76.1) | 36985 (95.1) | 16892 (88.8) | 11863 (52.9) | 11265 (53.8) | |
| Vaccinated < 6 months | 10271 (10.1) | 0 (0.0) | 871 (4.6) | 6337 (28.3) | 3063 (14.6) | |
| Vaccinated > 6 months | 3879 (3.8) | 0 (0.0) | 0 (0.0) | 1042 (4.6) | 2837 (13.5) | |
| Contact age category (%) | | | | | | 9.9 |
| 18–65 | 63671 (63.2) | 27275 (70.3) | 11230 (59.1) | 13228 (59.1) | 11938 (58.2) | |
| 0–18 | 31798 (31.6) | 9305 (24.0) | 6662 (35.1) | 7912 (35.3) | 7919 (38.6) | |
| 65+ | 5255 (5.2) | 2243 (5.8) | 1110 (5.8) | 1253 (5.6) | 649 (3.2) | |
| Number of tests last 3 months (%) | | | | | | 0.0 |
| 0 | 22031 (19.7) | 7249 (17.1) | 4289 (21.1) | 5249 (19.3) | 5244 (24.0) | |
| 1 | 80883 (72.4) | 33891 (80.1) | 14580 (71.8) | 19307 (70.8) | 13105 (60.1) | |
| 2+ | 8760 (7.8) | 1155 (2.7) | 1442 (7.1) | 2704 (9.9) | 3459 (15.9) | |
| Number of contacts who performed a test during the 10 days following their contact with the index (%) | 46417 (41.6) | 13392 (31.7) | 12808 (63.1) | 11043 (40.5) | 9174 (42.1) | 0.0 |
| Number of contacts who died from COVID-19 (%) | 130 (0.1) | 20 (0.1) | 18 (0.1) | 86 (0.2) | 6 (0.03) | 0.0 |

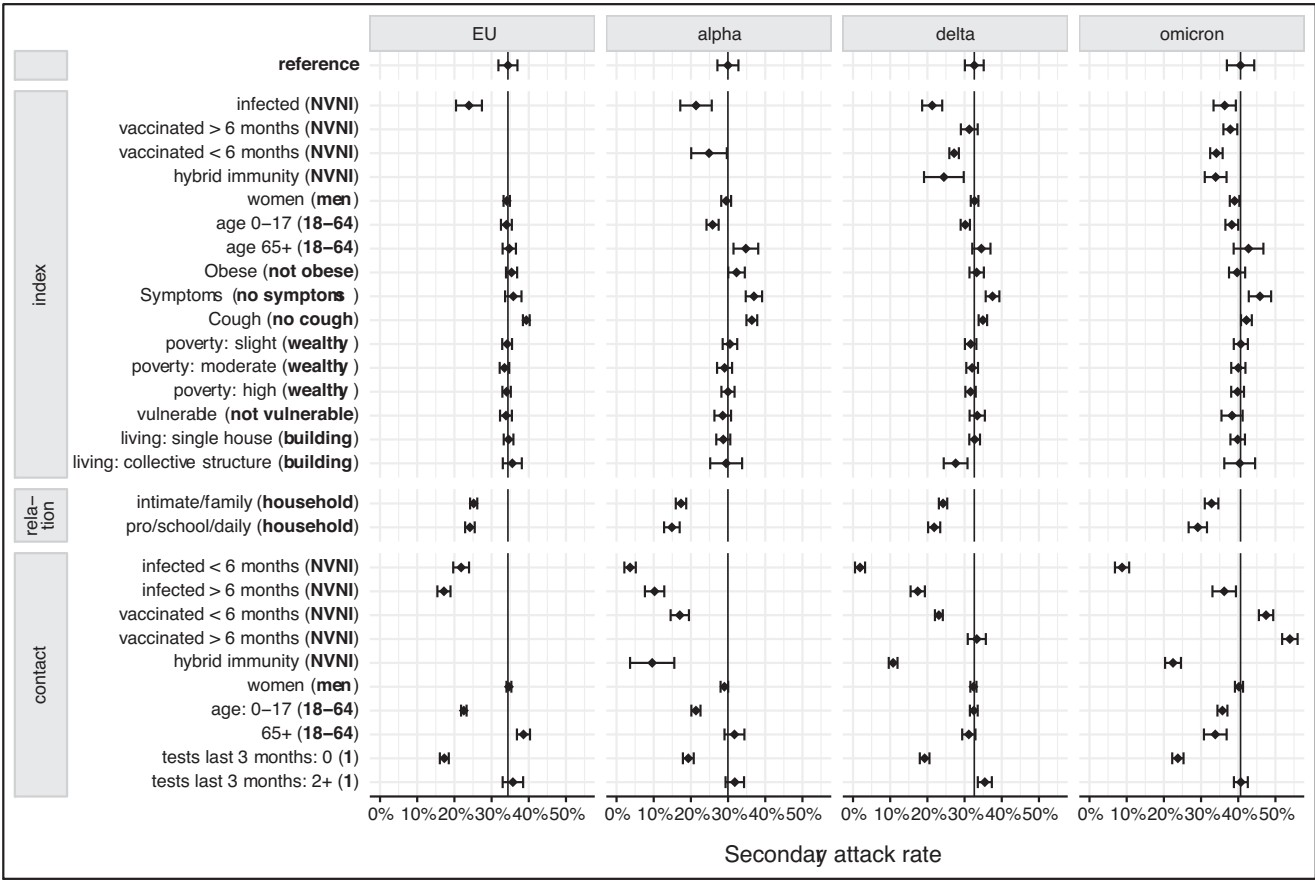

**Fig. 1 | Adjusted secondary attack rate.** Estimated Adjusted Secondary Attack Rate (diamond) and its 95% confidence interval (error bars) stratified per variant (EU, alpha, delta and omicron), with the reference value indicated with a vertical line (reference of each covariate is indicated in bold in parenthesis). Estimates and confidence intervals are produced by a generalized estimating equations linear regression with robust standard errors predicting a binary outcome indicating if the contact was infected by the index or not, using the index cases as cluster and an exchangeable correlation structure. The estimates are based on 42,295 index-contact relations for the EU variant, 20,311 for the alpha variant, 27,260 for the delta variant and 21,808 for the omicron variant. Estimates are adjusted for the index case gender, age, obesity, presence of symptoms, presence of cough, immunity before the index-contact date (SAR change of −5.1pp [−9.9, −0.3] during alpha, −5.4pp [−6.7, −4.1] during delta and −6.5pp [−8.2, −4.8] during omicron). A small but significant protective effect of vaccination was observed during omicron (−2.7pp [−4.6, −0.9]) when performed more than 6 months before getting infected. The recent vaccination of contacts had a strong protective effect for alpha (−12.9pp [−15.4, −10.5]) and delta variants (−9.5pp [−10.6, −8.5]). In this multivariable model without interaction, recent contact vaccination increased the SAR during the omicron wave. This increase vanished when considering an interaction between the immune status and the number of tests performed the last 3 months (2.4pp [−1.6, 6.4] and 3.0pp [−1.9, 7.8] if the contact performed 1 or more than 2 tests, respectively). If vaccination occurred more than 6 months before the last meeting between index and contact, it did not have a significant effect during the delta variant and even had a net tendency to increase the SAR with Omicron (increase of 13.3pp [11.2, 15.3]). This increase remained similar even when considering an interaction between the immune status and the number of tests performed 3 months before. Hybrid immunity had on both the susceptibility to get infected and on the infectiousness a higher effect than vaccination but lower than recent infection. It indeed decreased the SAR of 20.4pp [14.3, 26.3], 21.8pp [20.6, 23.0], and 18.2p [16.0, 20.3] for the contact during the

status, neighbourhood socioeconomic condition, vulnerability and type of living; the link between the index case and its contacts, and for the contact persons, their gender, age, number of tests performed the three months before the contact date with the index case, and their immunity status. The reference index case–contact relation of this multivariate analysis is the contact between two men of age below 65 living at the same place, the index being not vaccinated not infected (NVNI), not obese, living in a wealthy neighbourhood and being not a vulnerable person, living in a housing building, and the contact person being a NVNI adult men who performed one SARS-CoV-2 test during the last 3 month preceding the contact. Exact values of the estimated can be found in Table 2, and unadjusted estimates are presented in supplementary table S4.

alpha, delta and omicron waves respectively, and of 8.2 pp [2.8, 13.5] and 6.7pp [3.8, 9.6] for the index during the delta and omicron wave. The combined recent vaccination for both contact and index (interaction between both immune status) decreased the SAR by 22pp [13, 32] during alpha and 16pp [14, 19] during delta, but had no significant effect during omicron.

Of note, the effect of immunity on SARS-CoV-2 propagation is shared with a 1:3 ratio between the reduction of infectiousness and the reduction of infection susceptibility (see Fig. 2) for previous infections, but not for recent vaccination. Indeed, this ratio seems to lower with new variants, and even reversed for Omicron, where recent vaccination has no effect anymore on susceptibility but an increased effect on infectiousness.

**Effect of testing during the past 90 days**
The propensity of contacts to perform tests, measured by the number of tests performed by the contacts the last 90 days preceding their last encounter with the index, had a large effect on SAR calculation. Those who did not perform any test during this period had a reduced SAR of 17.1pp [15.9, 18.3], 10.6pp [9.2, 12.1], 13.4pp [12.1, 14.7] and 16.9pp [15.3, 18.4] for the EU1, Alpha, delta and omicron respectively when compared to those who performed one test. Performing two tests or more

**Table 2 | Estimated coefficients of the multivariable generalized estimating equation [Confidence Interval], providing the additional effect of each variable on the reference secondary attack rate (first line), for the 4 periods of dominance of the variants EU1, alpha, delta, and omicron**

| | | EU1 | Alpha | Delta | Omicron |
|---|---|---|---|---|---|
| | Reference | 34.4*** [31.8,37.0] | 29.9*** [27.1,32.7] | 32.6*** [30.1,35.1] | 40.6*** [36.9,44.3] |
| Index | Immunity: previously infected (**NVNI**) | −10.5*** [−14.0,−7.0] | −8.6*** [−12.8,−4.3] | −11.3*** [−14.0,−8.6] | −4.3** [−7.3,−1.3] |
| Index | Immunity: vaccinated < 6 months (**NVNI**) | | | −1.3 [−3.6,0.9] | −2.7** [−4.6,−0.9] |
| Index | Immunity: vaccinated > 6 months (**NVNI**) | | −5.1* [−9.9,−0.3] | −5.4*** [−6.7,−4.1] | −6.5*** [−8.2,−4.8] |
| Index | Hybrid immunity (**NVNI**) | | 20.3 [−13.0,53.5] | −8.2** [−13.5,−2.8] | −6.7*** [−9.6,−3.8] |
| Index | women (**men**) | −0.3 [−1.2,0.5] | −0.5 [−1.8,0.9] | 0.1 [−0.9,1.1] | −1.6* [−2.9,−0.3] |
| Index | age 0–17 (**18–64**) | −0.4 [−1.9,1.0] | −4.1*** [−5.8,−2.5] | −2.4*** [−3.7,−1.2] | −2.3** [−4.0,−0.6] |
| Index | age 65+ (**18–64**) | 0.4 [−1.4,2.1] | 4.8** [1.5,8.1] | 1.9 [−0.6,4.3] | 2.2 [−1.8,6.2] |
| Index | Obese (**not obese**) | 1.0 [−0.5,2.5] | 2.3* [0.1,4.6] | 0.7 [−1.3,2.6] | −1.0 [−3.1,1.2] |
| Index | Symptoms (**no symptoms**) | 1.4 [−0.8,3.6] | 7.0*** [4.8,9.2] | 4.9*** [3.1,6.7] | 5.2*** [2.2,8.2] |
| index | Cough (**no cough**) | 4.9*** [4.0,5.8] | 6.4*** [5.0,7.9] | 2.3*** [1.2,3.5] | 1.6* [0.1,3.0] |
| index | neighbourhood poverty: slight (**wealthy**) | −0.2 [−1.5,1.1] | 0.6 [−1.4,2.5] | −1.0 [−2.5,0.6] | 0.1 [−1.8,2.0] |
| index | neighbourhood poverty: moderate (**wealthy**) | −0.9 [−2.2,0.4] | −0.9 [−2.9,1.1] | −0.6 [−2.1,1.0] | −0.6 [−2.5,1.3] |
| index | neighbourhood poverty: high (**wealthy**) | −0.4 [−1.5,0.8] | 0.1 [−1.7,1.8] | −1.0 [−2.4,0.4] | −0.8 [−2.5,0.9] |
| index | vulnerable (**not vulnerable**) | −0.5 [−2.1,1.1] | −1.4 [−3.6,0.9] | 0.8 [−1.3,2.8] | −2.4 [−5.2,0.5] |
| index | living: single house (**building**) | 0.2 [−1.1,1.5] | −1.2 [−3.1,0.7] | 0.1 [−1.3,1.5] | −0.7 [−2.7,1.2] |
| index | living: collective structure (**building**) | 1.2 [−1.4,3.7] | −0.5 [−4.8,3.8] | −5.0** [−8.2,−1.8] | −0.1 [−4.3,4.0] |
| Index - Contact | intimate/family (**housing**) | −9.2*** [−10.2,−8.2] | −12.6*** [−14.0,−11.2] | −8.4*** [−9.5,−7.3] | −7.8*** [−9.6,−5.9] |
| Index - Contact | pro/school/daily (**housing**) | −10.2*** [−11.5,−8.9] | −15.1*** [−17.2,−13.0] | −10.8*** [−12.4,−9.1] | −11.4*** [−13.9,−9.0] |
| Contact | Immunity: previously infected <6 months (**NVNI**) | −12.6*** [−14.7,−10.5] | −26.3*** [−27.9,−24.8] | −30.7*** [−32.1,−29.4] | −31.9*** [−33.8,−30.0] |
| Contact | Immunity: previously infected > 6 months (**NVNI**) | −17.2*** [−19.0,−15.5] | −19.7*** [−22.3,−17.1] | −15.2*** [−17.1,−13.3] | −4.4** [−7.6,−1.2] |
| Contact | Immunity: vaccinated <6 months (**NVNI**) | | −12.9*** [−15.4,−10.5] | −9.5*** [−10.6,−8.5] | 6.9*** [4.9,8.8] |
| Contact | Immunity: vaccinated > 6 months (**NVNI**) | | | 0.7 [−1.8,3.1] | 13.3*** [11.2,15.3] |
| Contact | Immunity: hybrid (**NVNI**) | | −20.4*** [−26.3,−14.4] | −21.8*** [−23.0,−20.6] | −18.2*** [−20.3,−16.0] |
| Contact | women (**men**) | 0.3 [−0.4,0.9] | −1.0 [−2.0,0.1] | −0.2 [−1.1,0.6] | −0.4 [−1.5,0.7] |
| Contact | age: 0–17 (**18–64**) | −11.9*** [−12.6,−11.1] | −8.6*** [−9.8,−7.4] | −0.1 [−1.2,0.9] | −4.9*** [−6.2,−3.6] |
| Contact | 65+ (**18–64**) | 4.2*** [2.4,5.9] | 1.8 [−0.9,4.4] | −1.5 [−3.2,0.3] | −6.8*** [−9.8,−3.7] |
| Contact | Number of tests last 90 days: 0 (**1**) | −17.1*** [−18.3,−15.9] | −10.6*** [−12.1,−9.2] | −13.4*** [−14.7,−12.1] | −16.9*** [−18.4,−15.3] |
| Contact | Number of tests last 90 days: 2+ (**1**) | 1.3 [−1.4,4.1] | 1.9 [−0.6,4.4] | 2.8** [0.9,4.7] | 0.1 [−1.8,2.0] |

Estimates, *p* values and confidence intervals are produced by a generalized estimating equations linear regression with robust standard errors predicting a binary outcome indicating if the contact was infected by the index or not, using the index cases as cluster and an exchangeable correlation structure. Estimates are adjusted for the index case gender, age, obesity, presence of symptoms, presence of cough, immunity status, neighbourhood socioeconomic condition, vulnerability and type of living; the link between the index case and its contacts, and for the contact persons, their gender, age, number of tests performed the three months before the contact date with the index case, and their immunity status. The reference index case–contact relation of this multivariate analysis is the contact between two men of age below 65 living at the same place, the index being not vaccinated not infected (NVNI), not obese, living in a wealthy neighbourhood and being not a vulnerable person, living in a housing building, and the contact person being a NVNI adult men who performed one SARS-CoV-2 test during the last 3 month preceding the contact. The reference category for each categorical variable is indicated in bold in parenthesis. The left column indicates if the variable concerns the index case, the contact, or their relation. p values are indicated with *. *: 0.01 < *p* < 0.05, **:0.001 < *p* < 0.01, ***: *p* < 0.001

did not clearly increase the SAR (an increase of 3.0pp [1.1, 4.8] only during the delta wave). The propensity of contacts to perform tests modified the effect of immunity on SAR when comparing univariable and multivariable adjustment (these results are detailed in supplementary material, see supplementary table S4). After adjustment for an interaction between contact immunity of the number of tests performed by contact during the past 90 days, testing more enhanced the effect of protection against infection conferred by recent infection and hybrid vaccination. However, it decreased this effect for other immune status, especially during the omicron wave (see supplementary figure S4 and supplementary table S2).

### Other characteristics

When the contacts between index and contact took place outside the household, either with family or intimate partner, or professional/recreational setting, the SAR was substantially lower (around 10 pp for all VoC). Cough increased the infectiousness, but with an amplitude decreasing with the new variants. A coughing index increased the SAR by 4.9pp [4.0, 5.8] and 6.4pp [5.0, 7.9] for the EU1 and alpha variant, but this effect was reduced to 2.3pp [1.2, 3.5] for the delta variant and 1.6pp [0.1, 3.0] for omicron variant. Contact children had a lower SAR, especially for early variant. Contact older than 65 tended to be more infected at the beginning of the pandemic, but this effect became non-significant for alpha and delta, and even reversed for omicron, with contact older than 65 years having a SAR 6.8pp [3.7, 9.8] lower than the reference category. Concerning the age of the index, index children seemed to be slightly less contagious than adults. Being an infected adult older than 65 years increased infectiousness only during the alpha wave (4.8pp [1.5, 8.2]).

When stratifying for gender, multivariable models showed similar pattern of results (see supplementary figure S5a, b).

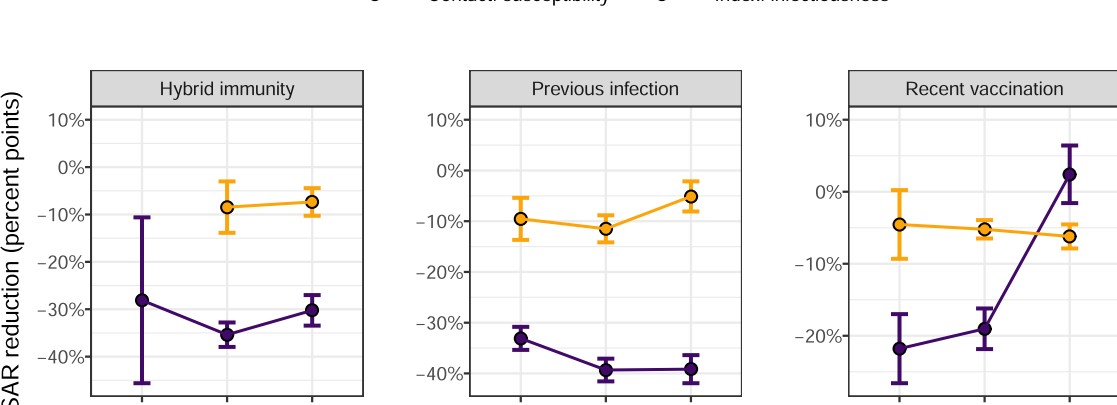

**Fig. 2 | Immunity, susceptibility to be infected and infectiousness.** Effect of immunity (recent vaccination, recent infection or hybrid immunity) on the susceptibility to be infected (magenta) or on the infectiousness (yellow), expressed as the estimated percent point change of secondary attack rate (circle) and its 95% confidence interval (error bars), stratified per period of variant predominance. Estimates and confidence intervals are produced by a generalized estimating equations linear regression with robust standard errors predicting a binary outcome indicating if the contact was infected by the index or not, using the index cases as cluster and an exchangeable correlation structure. Estimates are adjusted for the index case gender, age, obesity, presence of symptoms, presence of cough, immunity status, neighbourhood socioeconomic condition, vulnerability and type of living; the link between the index case and its contacts, and for the contact persons, their gender, age, number of tests performed the three months before the contact date with the index case, their immunity status, and an interaction between immunity status and number of tests performed. The reference index case–contact relation of this multivariate analysis is the contact between two men of age below 65 living at the same place, the index being not vaccinated not infected (NVNI), not obese, living in a wealthy neighbourhood and being not a vulnerable person, living in a housing building, and the contact person being a NVNI adult men who performed one SARS-CoV-2 test during the last 3 month preceding the contact. Same results are presented graphically in Fig. 1.

## Sensitivity analysis

Six sensitivity analyses were conducted to assess the influence on the results of: the imputation methods used (results of the multivariable analysis in supplementary figure S6 and supplementary table S5, with the interaction term presented in figure S7 and table S6), the choice of the variables to adjust for the contacts' propensity to test (descriptive statistics of the alternative variable is provided in supplementary table S7, the results of the multivariable analysis in supplementary figure S8 and supplementary table S8, with the interaction term presented in figure S9 and table S9), the definition of the variant periods (see supplementary figure S10 and table S10), and the potential contact-case misclassifications (misclassification of community cases in figure S11 and table S11, misclassification of primary case S12 and table S12 and misclassification of tertiary case supplementary figure S13 and table S13). Results of these analyses can be found in supplementary material. These analyses yielded similar results to the analyses presented above.

## Discussion

In this study of >50,000 index cases and >110,000 declared contacts, spanning four different SARS-CoV-2 variants circulating over almost 2 years, we observe that the immunity conferred by vaccine or infection lowers both the infectiousness and the susceptibility to infection, and that a previous infection contributes more to the reduction of the virus propagation. The main immune factor lowering the secondary attack rate was natural infection, while vaccination had a more limited impact, even when recent enough. The reduction of infectiousness conferred by vaccination appears to wane less in time and to be less sensitive to variant changes than the decrease of infection susceptibility, making this effect the major contribution of vaccination to the reduction of SARS-CoV-2 propagation for Omicron. The other variables affecting the transmission of SARS-CoV-2 were the age of the contact person, the presence of symptoms - especially cough - for the index, the setting of the encounter between index and contact (e.g., home, work) and the tendency of the contact to get tested.

Compared to non-vaccinated and never-infected person, vaccination was protective for both index and contacts, the effect for the index being smaller than for contacts, as reported previously[36]. Because of the waning of the vaccine-induced immunity[10] and the immune escape of successive variants when compared to the previous ones[8,37], the timing of the last vaccination and the VoC concerned were important, especially for the contact. Vaccination, even when performed less than 6 months before, did not add any protection to contacts during the omicron wave. On the other hand, the escaping capacities of omicron did not affect the reduction of infectiousness conferred by recent vaccination. This suggests that vaccine still lowers the viral load of persons infected by Omicron, in agreement with the fact that vaccine diminishes the occurrence of severe disease for this VoC[13,38]. Although vaccination more than 6 months ago still had some effect on the infectiousness of index cases, it added no protection against infection for the contacts during delta wave, and even had an opposite effect (i.e. an increase of the SAR) during the omicron wave. This counterintuitive effect might be due to a combination of the strong immune escape of this variant[39] and the tendency of vaccinated people to comply less with COVID-19 mitigation strategies[40], such as physical distancing and mask recommendations. Such result has been observed in a previous study[41]. Infected unvaccinated indexes had a reduced SAR across all variants. This reduction was higher than the one observed for vaccination for Delta, but not for Omicron, in agreement with recent measurement of viral load dynamics[24]. Previous infection also showed a strong protective effect against being infected for the contacts, even more after adjusting for their tendency to test. This protection is reduced after 6 months. The waning of this protection conferred by previous infection is rather small for early variants, in line with the recently observed slower immunity waning after infection when compared to vaccine[7,16,18], but is substantial for Delta and Omicron, due to their stronger potential for immune escape[42]. This protection against infection was higher than vaccination for all variants (up to 7 times higher for Omicron, in agreement with recent estimate of Gazit and co-authors[17]). This higher and longer lasting protection of the infection when compared to vaccine-induced immunity may find its root in a more global immune response and may be due to specific IgA response[43]. Hybrid immunity provided stronger protection and reduction of infectiousness than vaccines, as observed elsewhere[44],

but no higher than recent infection, in agreement with a large Israeli study[18].

The association of SAR and immune status, either due to recent vaccine or previous infection, changed notably between univariable and adjusted analyses. Though previous infection and recent vaccination were protective in univariable models, it became more so, for all VoCs, in the multivariable model. The main confounder of this association was the tendency of the contact to test, which modified the SAR of the non-immune population, our reference category. Indeed, the SAR was much higher among non-immune people who tested compared with those who did not, because they were more likely to test after being exposed to the index case. But the SAR was quite similar among previously infected or recently vaccinated people, irrespective of their tendency to test, suggesting that the protective effect of immunity was stronger than the tendency to test.

It is of note that the adjustment in our analysis corrects strongly the SAR value of each variant, resulting in a similar value for the EU1, alpha and delta variant, but lead to a higher SAR for omicron, similar to what has been reported by large reviews[20]. The context of the encounter between the index and the contacts greatly affected the SAR, with more distant relationships (work, leisure) leading to lower SAR than housing relation, as noticed elsewhere[29]. The policy implication of this finding could be that requiring a quarantine only of household contacts instead of all contacts is an appropriate solution to reduce the burden of health policy without increasing significantly the transmission.

Symptomatic indexes have consistently been shown to increase SAR since the beginning of the pandemic[30,31]. However, the difference in SAR between symptomatic and asymptomatic is relatively small, suggesting that everyone should be careful to minimize their risk of transmitting the disease, even if not symptomatic. With respect to coughing, the impact of coughing, though significant, decreased for later VoCs. This could be due to a combination of a higher adherence to mask-wearing within the population during these periods, and of changes in infection pathway. Indeed, since omicron infects mostly upper respiratory tract[45] and produces a higher viral load[46], the higher quantity of virus expelled when naturally breathing or sneezing could explain the lower comparative effect of coughing for this particular VoC.

As shown in previous studies[31], we also found that contact children had a lower SAR (both as index and contact) than adults. It has been postulated that difference in contact type, quantity of virus expelled, decreased receptor expression in the respiratory tract or age-related increase in innate immune response in children could explain this difference[47–49], but the tendency of children to be more asymptomatic[50] could also play a role, as they tend to be less tested. However, this difference with adults decreased with delta and omicron variants compared with the other VoCs. This change is potentially due to both the preference of the new variants for this more immune and unvaccinated population[51] and to a potential detection bias (children tended to be less tested at the beginning of the pandemic in Geneva).

Adults older than 65 years had a slightly higher adjusted SAR during the early waves, as reported elsewhere[47]. This effect disappeared later in time, probably due to multiple factors, such as the implementation of physical distancing and protection, but also detection bias. The above-mentioned underrepresentation of this population in this study could also bias this result.

Interestingly, we found no association between living or personal socio-economic circumstances (SEC) and SAR. This result is in line with what was reported by a recent seroprevalence study in Geneva[52]. It has to be noted that our study does not concern the first wave of SARS-CoV-2 pandemic, and the association between COVID-19 variables and socioeconomic condition vary greatly among waves[53]. Disparities across the social ladder of the society concerning COVID-19 have been shown to concern mainly the access to test[53–55], and the COVID-19

mortality and morbidity[56,57]. Even if the dependence of SAR on some socio-economic variables have been shown in small samples in some countries[58,59], it may be dependent on a particular situation or time.

The associations between all variables and SAR were mostly similar between men and women, in agreement with seroprevalence studies in Switzerland[51,60,61] and other SAR studies[31], which have shown that gender or sex affects access to healthcare, morbidity and mortality, but not the infectiousness of SARS-CoV-2. Nevertheless, previous infections of women index were associated with lower SAR for the EU1 and alpha variants.

The strength of this study is based on the operational database gathering all SARS-CoV-2 tests performed by a large population of indexes and their contacts, covering 2 years of pandemics and multiple variants of concern. The availability of detailed information on cases and contacts together with the high number of measures allowed adjusting for a wide range of covariates while keeping a high number of points per event. In particular, the availability of vaccination status, for both index and contact, adds to the strength of the study. Finally, the canton of Geneva invested a lot of effort in testing and following vulnerable populations during the pandemics, including undocumented migrants, thereby reducing potential selection bias.

This study also has limitations. As previously mentioned, people over 65 years old are underrepresented, while young people are overrepresented. The underrepresentation of old people may be due to the handling of contact tracing and isolation by their specific nursing home or healthcare facility, or due to their health status which did not allow them to provide their contacts. As a consequence, vaccinated people, who tend to be older, were also underrepresented in our cohort. This could potentially lead to selection bias. Nevertheless, the case fatality rate of the index who reported contacts (0.4%) corresponds to the overall case fatality rate in the register, indicating that the severity of the cases reporting contacts seem similar to the overall COVID-19 positive population. The main limitation of this observational registry study is information and surveillance bias[62], including measurement error. There are several sources of these bias. First, during peak epidemics, most contact declaration were made using a self-complete online form, and less contacts were declared. Though this could be due to a real reduction in number of persons seen during peak epidemics, it could reflect information bias, due to the fact that self-complete forms do not elicit as many contacts as oral interviews with subsequent questions on potential contacts. This could modify the way the index recalled their contacts and cause potential bias, which should be mitigated by adjusting for cases and contact characteristics. A second source of information bias is the dependence of the attack rate estimation on the tests being performed, since contacts will be considered positive only if they were tested. This affect both our immunity categories (some persons are classified as non-infected whereas they actually are infected) and our outcome (some contacts are infected but not tested). In the first case, this would underestimate the large difference observed between non-infected person and other immune categories. In the second case, this would underestimate the SAR, and could lead to residual confounding. In this study, we found that the tendency to get tested (number of tests in the 90 days before the date of contact) strongly influenced SAR, with people not testing in the preceding months having a much lower SAR. This propensity of the population to be tested, and the delay between tests and health authority action, varies over time and depends on the health policies implemented. The change in testing is especially visible for children, for whom the testing policies varied from almost no tests during the first waves, even when they were contacts (in part due to recommendations[63] but also because they are often not symptomatic[50]) to compulsory autogenic testing in schools if more than two children were infected in a classroom by the end of 2021. The influence of health policies changes can be seen in the increasing delay between the index test result and the first call to contacts with

advancing pandemics. Similarly, the introduction of the Swiss sanitary pass the 26th of June 2021 affected the testing of the population. Indeed, since December 2021 it allowed vaccinated or previously infected patients to use common social venues when non-vaccinated and non-infected persons needed a negative test to do so. Although the adjustment for the propensity to test and for its interaction with the immune status confirmed and even strengthened the effect of immune status on SAR, we cannot completely rule out residual bias, inherent to any observational study. Finally, reduced testing may occur among people from low socio-economic conditions, in order to escape quarantine[53]. Again, despite adjusting for the propensity of the contact to test, there can be some residual confounding partly explaining the absence of association between SEC and SAR. A third source of information bias is the supposition that the index cases are the primary cases, and that the contacts becoming positive in less than 10 days after their last contact with the index cases are the secondary cases. Though we cannot rule out misclassification (some contacts may actually be the primary cases or tertiary case), the three sensitivity analyses performed to address potential misclassifications indicate that our results are robust to this measurement error.

As in any registry study, we cannot rule out residual confounding. Though extensive, the adjustment certainly misses some potential confounders. In addition, the categorization performed on variables such as contact type or age, or the behaviours changes associated with vaccination, could still lead to some confounding.

Lastly, the study did not assess variant by genotype results based on a PCR test but was based on period of time of variant dominance. Due to an overlap between every variant change, this could alter our results, but probably in a minimal way since variants became dominant quite quickly after they emerged. In addition, though this stratification can be considered a strength since it accounts for differences in vaccination effect across variants, it inflates type I error by using four models without correcting for multiple tests.

Our study shows that mRNA vaccination alone, although effective for reducing severe outcomes or hospitalisations[13,38], had a limited effect but is not sufficient anymore to contains or moderate SARS-CoV-2 propagation. Infections have important reduction effect on the virus transmission but they are associated, apart from the known risks of an acute infection, with cumulative long term effects of SARS-CoV-2 infections[64] provoking potential immunity deficiency[65], long lasting symptoms[66], including cardiac[67] and neurological[68] damages. These health consequences concerning an increasing part of the population[69], public health policies should intend reducing the number of infections for all persons, vaccinated or not, with effective and socially acceptable nonpharmaceutical interventions such as air purification[70,71], ventilation[72,73], or mask wearing[74]. Finally, to be able to study the evolution of the SARS-CoV-2 among the population, it is of prime importance to continue to monitor the infections in the community and the general population[75].

## Methods

### Setting and period
Data used for the present study consisted in a register dataset of links between an infected case (hereafter the index case) and a declared person with whom he/she had close contact during the 10 days preceding his/her test result (hereafter the contacts). These data stem from the ARGOS database[35], which is an ongoing operational COVID-19 database created by the Geneva health state agency (Geneva Directorate of Health), based on the REDCap software[76].

Geneva is a mainly urban state of 511,921 inhabitants as of the last census in December 2021[77], with a high population density. It doubles its population on working days (excluding pandemic restrictions) as a result of national and international commuter traffic (mainly from neighbouring France). We used data from the 26th February of 2020 (first positive tested recorded in Geneva) to 28th February of 2022.

Data was not collected after March 1st 2022 because contact declaration was stopped at this date in Switzerland. Details about the number of cases, the number of tests and the number of COVID-19-associated death during the period of interest in Geneva (supplementary figure S14) as well as a description of the main non-pharmaceutical interventions (supplementary figure S15) are provided in supplementary material. This research has not been restricted and received the agreement of the Cantonal Ethic Committee of Geneva (CCER protocol 2020-01273). Participants had the possibility to refuse sharing their data for research through a form that was automatically sent. Those who did were removed from the analysis. The research included local researchers throughout the research process. Role and responsibilities were agreed amongst collaborators. The research is locally relevant and had been determined with the local health institutions. Local and regional research relevant to our study has been taken into account. The research conducted does not result in stigmatization, incrimination, discrimination or otherwise personal risk to participants, and did not involve health, safety, security or other risk to researchers.

### Index cases and contacts
The ARGOS register contains baseline, follow-up, and contact information of all SARS-CoV-2 positive tested persons (index case) residing in the State of Geneva, Switzerland.

A contact was considered as infected by the index case if they had a positive COVID-19 result within 10 days following their last contact with the index case. Declared contact in Geneva had the obligation to quarantine during 10 days since the implementation of contact tracing, except for children below 12 years. The 8th February, 2021, it was allowed to shorten the quarantine at day 7 with a negative SarS-CoV-2 PCR test. The quarantine was later shortened to 7 days (31st of December 2021) and to 5 days (12th of January 2022). By end of 2021, vaccinated persons or persons with a positive test during the last 4 months did not have the obligation to quarantine after a contact with an infected index. Since October 2020, health professionals were allowed to work even if quarantined. From February 2020 to end of April 2020, contact information was collected by interviewing the index case. From May 2020, index cases had the possibility to provide their contacts names and phone through an online form. Contacts were then approached using phone interviews. Additionally, an online form was implemented at the end of September 2020 to support the oral interviews, allowing the contacts to complete the required information themselves right after receiving the notification of their positive test. The interview form and the online form were identical. From mid-December 2021, the oral interviews could not be maintained, therefore contact information was only gathered from the online formula.

Contact information contained the type of contact setting between the index case and the declared contacts (see supplementary material), the date of the last contact between index and contact, the birth date, gender, date of subsequent or anterior positive PCR or antigenic test results, as well as the living address and the vaccination dates. Information about the index cases included date of SARS-CoV-2 test result, gender, date of birth, living address, symptoms (see list of symptoms in supplementary material), personal vulnerability variables, vaccination dates and date of previous infections. In the present study, the dataset is composed of the index case and contact dyads residing in Geneva. An index case could appear for various infections, and a contact could appear with various index cases.

### Secondary attack rate (outcome)
The secondary attack rate[78], first described by Dr Chapin at the beginning of the last century, refer to the probability of infection among close contacts of an index case in a particular setting (work, household, …)[79] and is one of the key estimate of the transmissibility of the virus. Its raw estimation consists in dividing the number of infected

contacts by the total number of susceptible contacts declared by the index cases. Adjusted estimation of SAR can be performed using linear regression methods (see statistical analysis).

### Immunity status (main predictor)

Immunity status was calculated at the date of the last contact between the index and the contact and was categorized in the following categories:

- Vaccinated more than 6 months or less than 6 months. This category included all persons with at least one dose of a vaccine recognized in Geneva, including booster doses, for which the last date of vaccination was more or less than 6 months.
- Infected at least one time, more than 6 months or less than 6 months. This category included persons not vaccinated but having at least one positive PCR test result, more or less than 6 months ago.
- Not vaccinated not infected (NVNI). This category included persons not vaccinated and not infected previously to the date of last contact between index case and the contact.
- Hybrid infections: persons with complete vaccine scheme and previous infection.

The most administrated vaccine type in Geneva was the mRNA-based vaccines, such as Moderna mRNA-1273 (59.83% of the total administrated vaccine doses in Geneva) and Pfizer BNT162b2 (39.86%). Other type of vaccine stands for a minor part: Janssen (0.25%) and Nuvaxovid (0.06%). The final uptake at end of February 2022 was of 71.6% of the population who received at least one dose of vaccine (see supplementary figure S16). Details on the vaccination roll-out in Geneva can be found in supplementary materials. More than 95% of vaccinated people had a complete vaccination scheme.

### Controls

The estimation of the effect of immunity of both contact and index on SAR was controlled by the age and gender of the index and contact, the body mass index (BMI) of the index, the presence of symptoms and cough for the index, the type of building in which the index is living, the neighbourhood socioeconomic condition of the index, personal vulnerability of the index, the type of relation between index and contact, and the propensity of the contact to test.

Three groups of age were used to categorize people: 0–17 years, 18–64 years and above 65 years (65 + ). BMI was calculated from height and weight and was categorized in obese and non-obese categories. For age superior to 18 years, obese was considered for BMI above 30 kg/m². For age below 18, we used the extended international body mass index cut-offs corresponding to the threshold of 30 kg/m2 at 18 years old[80]. Presence or absence of symptoms was operationalized as 1 if the person reported any symptoms, otherwise 0. Cough was defined as the presence of dry or wet cough symptoms. Categorization of the socio-economic condition of the neighbourhood area (417 official neighbourhood areas in the State of Geneva) was, similarly to previous work[53], based on an index provided by the centre for the analysis of territorial inequalities (see supplementary materials). The statistical office of Geneva provided the type of building and number of inhabitants for each address. Addresses were geo-coded using the exhaustive list of all addresses of the State of Geneva. The building type were categorised in three categories: building (multi-residential building, potentially having shops), single houses, or collective structure. This last category included nursing homes, jails, asylums and fire-stations. A person was considered vulnerable if the person reported difficulty to make ends meet, lived in a highly subsidized housing, or if they asked explicitly to avoid police control. The type of relationship between index and contact was operationalized in three categories: living under the same roof, having an intimate or familial relationship (but not living under the same roof), or other relationship. The

correspondence between the initial categories available in the dataset and the three categories for the present study is described in the supplementary material. Tendency to test was estimated by counting the number of tests performed by each contact during the last 3 months preceding their encounter with the index case. This number was categorized in three categories: 0, 1 and more than 2 (2 + ). In a sensitivity analysis, we also considered the number of tests performed in the last 6 months, categorized into four categories: 0, 1, 2 and more than 3 (3 + ), see sensitivity analysis subsection.

### SARS-CoV-2 variants

As the ARGOS data did not contain information about the SARS-CoV-2 variant type, we divided the study periods according to the predominance of the SARS-CoV-2 variant of interest, based on the data provided by the Global Initiative on Sharing All Influenza Data[81] for the Geneva region, which stem mainly from wastewater analysis. To do this, we modelled the evolution of the share of variants as rising and falling sigmoids (see supplementary figure S17) and determined the period of predominance with one VoC above 50%:

- EU1: from 01-06-2020 to 05-01-2021
- Alpha from 06-01-2021 to 14-06-2021
- Delta from 15-06-2021 to 17-12-2021
- Omicron from 18-12-2021 to 01-03-2022 (mainly BA.1)

In a sensitivity analysis we considered also periods defined with a threshold of 90% (see sensitivity analysis section).

### Statistical analysis

SAR was estimated using generalized estimating equations predicting a binary outcome indicating if the contact was infected by the index or not. The clusters considered were the index cases, and we assumed an exchangeable correlation structure. We used a Gaussian identity link[82], which allows to estimate the relative proportion increase provoked by each covariate relative to a reference proportion of infected contacts, that is the SAR. The independent variables of the regression were the immune status of the index case and its contact, and the control variables were the age and gender of the index and contact, the body mass index (BMI) of the index, the presence of symptoms and cough for the index, the type of building in which the index lives, the neighbourhood socio-economic condition of the index, personal vulnerability of the index, the type of relationship between index and contact, and the propensity of the contact to test.

Missing data were handled using multiple imputation with chained equations (20 samples, 5 iterations) at the person, infection or contact level (see supplementary materials). The analysis was then performed independently on each imputed dataset, and the results were pooled according to the Rubin's rules.

All analysis has been performed using R 4.0.0[83], using the *geepack* library[84] for the general estimating equation, *mice*[85] for the multiple imputation with chained equation and *ggplot2* for the figures and graphs. The code used for the analysis-has been made available at the following Gitlab repository: https://gitlab.com/dmongin/scientific_articles/-/tree/main/Effect_of_mRNA_vaccination.

### Sensitivity analysis

We performed six sensitivity analysis, which can be found in the supplementary materials:

- The first sensitivity analysis concerned imputation. We performed the same analysis but with row wise complete case data. The resulting dataset has 48,468 lines.
- The second sensitivity analysis is concerned with the definition of the variant period. Instead of determining the variant period as the period during which the VoC was above 50% of the detected variant, we set a threshold of 90% of dominance. By doing so, we lose 18,812 measurements of our dataset.

- A third sensitivity analysis is concerned with the adjustment of the propensity of the contact to test. Instead of considering the number of tests during the 3 previous months, categorised in 0, 1 or 2 + ; we considered here the number of tests performed the last 6 months categorised in 4 categories: 0, 1, 2 and 3 + .
- The last three sensitivity analysis are concerned with potential misclassification of the contact cases[86]. We indeed considered in the main analysis the index case as the primary case. Three sensitivity analysis were conducted to address three different potential misclassifications:

  - Misclassification of community cases: the contact case is not infected by the index, but elsewhere in the community. In order to address this potential misclassification, we restricted our analysis to contact cases who were effectively placed in quarantine at maximum the day after their last contact with the index case (38,277 index-contact relations).
  - Misclassification of primary case: the index case is not the index case but the contact case, and the contact is the index case. To address this potential misclassification, we reproduced our analysis, but with a more restrictive definition of positivity for the contact case: we considered contacts as infected by the index only if their test result was at least 4 days after and still less than 10 days following their last contact with the index (resulting in 9,943 positive contacts, instead of the 21,435 of the initial dataset).
  - Misclassification of tertiary case: in case there are several contacts becoming positive for the same index case, there is the possibility that one of the contacts infect each other's. To address this potential misclassification, we restricted our analysis to households with only one contact (resulting in 8669 index-contact relations).

**Reporting summary**

Further information on research design is available in the Nature Portfolio Reporting Summary linked to this article.

## Data availability

Due to privacy issues, these individual-level Data are available upon request at https://edc.hcuge.ch/surveys/?s=TLT9EHE93C. Response is provided within two weeks. Data are provided de-identified and thus exact address is not available.

## Code availability

The code used for the analysis- has been made available at the following Gitlab repository: https://gitlab.com/dmongin/scientific_articles/-/tree/main/Effect_of_mRNA_vaccination.

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

## Acknowledgements

We thank the Geneva Directorate of Health for collecting and providing the data. This research was supported by the research project SELFISH, financed by the Swiss National Science Foundation, grant number 51NF40-160590 (LIVES Center international research project call) and attributed to D.S.C.

## Author contributions

D.M. conceived and designed the analysis, participated to data collection, performed data curation, performed the data analysis and participated to its interpretation, created the data visualizations, and wrote the paper. N.B. participated to the interpretation of the analysis and to the writing of the paper. G.L. participated to data collection and the data curation, participated to the interpretation of the analysis and revised critically the paper. G.S. participated to data collection and the data curation, participated to the interpretation of the analysis and revised critically the paper. D.L.V. participated to the interpretation of the analysis and to writing of the paper. S.C. participated to the interpretation of the analysis and to writing of the paper. D.S.C. managed funding and ethical authorizations, participated to the conception and design of the analysis, contributed to the data analysis interpretation, and participated to the writing of the paper. The Covid-SMC Study Group participated to the data collection and the design of the analysis. All contributors revised critically the paper and agreed to its final version. D.M., G.L., G.S. and D.S.C. had full access to the data underlying the study.

## Competing interests

The authors declare no competing interests.

## Additional information

## Covid-SMC Study Group

Lucienne Da Silva Mora[3], Lena Després[3], Rachel Dudouit[3], Béatrice Hirsch[3], Barbara Müller[3], Charlotte Roux[3], Géraldine Duc[3], Caroline Zahnd[3], Adriana Uribe Caparros[3], Guillaume Schimmel[3], Jean-Luc Falcone[3], Nuno M. Silva[3], Thomas Goeury[3], Christophe Charpilloz[3], Silas Adamou[3], Pauline Brindel[3], Roberta Petrucci[3], Andrea Allgöwer[3], Abdel Kadjangaba[3], Christopher Abo Loha[3], Emilie Macher[3], Marc Vassant[3], Nadia Donnat[3], Philippe Pittet[3], Dominique Joubert[3], Samia Carballido[3], Ariane Germain[3], Sophie Bontemps[3], Elisabeth Delaporte[3], Camille Genecand[3], Aliki Metsini[3], Valérie Creac'h[3], Virginie Calatraba[3], Laura Flüeli[3], Hippolyte Piccard[3], Dan Lebowitz[3], Aglaé Tardin[3] & Simon Regard[3]

