## [Peer Review File · Nature Communications]

Effect of SARS-CoV-2 prior infection and mRNA vaccination on contagiousness and susceptibility to infectionREVIEWER COMMENTS

Reviewer #1 (Remarks to the Author):

Summary

This was a cohort study examining factors associated with the secondary attack rate of four SARS-CoV-2 variants. The primary independent variable was immune status (immune naïve, prior infection, vaccination, hybrid immunity). Congratulations to the authors for conducting a difficult study with a large sample size.

The findings are valuable - most notably the adjusted estimates of secondary attack rates by immunity status (infection, vaccine, and hybrid immunity) for cases and contacts, demonstration of the value of the interaction term between immunity status and testing frequency (and its role in addressing confounding), and the list of variables independently associated with secondary attack rates.

These findings have relevance for understanding how to limit the future spread of SARS-CoV-2 variants and how to best time vaccinations.

I have several major and minor comments.

Major

Methods

1. Please can the authors add a summary of how long after being identified as a case the participants were contacted? (i.e., mean time to interview/form completion). This could provide insights into recall bias about their close contacts.
2. There was a lot of significance testing in the analysis. Please can the authors correct their estimates for multiple testing or justify why they have not done this correction and add this as a limitation to the discussion section?
3. Please can the authors describe whether the interview form matched the self-completion online form? If they differed, please describe the ways in which they differed and how the changes may have influenced the findings.
4. Did the authors have access to housing density metrics? This would be a very valuable type of metric to include in the analysis.

Results

5. Please can the authors conduct a sensitivity analysis using the complete data (without imputation) to compare to the main results given that there was a significant amount of missing data for some variables (e.g., contact type was 38% missing)? I recognize that this is a significant amount of work so perhaps they could do this for the key analysis only to evaluate the robustness of their imputation (key analysis being the multi-variate analysis with the interaction term between contact immunity and propensity of contact to perform tests)?

Discussion

6. Please can the authors briefly describe how switching from oral interviews to self-complete interviews may have impacted the results?
7. The authors may want to consider adding a limitation about selection bias. Is it possible that only moderately or mildly sick patients were able to respond to the questionnaire? If participants were very unwell (hospitalized) was it logistically possible for them to gain entry into the study? Is it possible that older people were under-represented because they were more likely to get severe disease or die and therefore not participate?
8. Was there any other severity information available in the registry? If all the participants that were very unwell or that died were excluded from the study, then it's possible that these results

only apply to persons with mild or moderate disease.

Minor

Page 3, Line 35-36: Consider deleting “Western countries” or changing it to “high income countries”. Boosters have been implemented for the purpose of maintaining immunity regardless of where they are being administered.

Page 4, line 62: The resulting dataset consisted in of 50’889 index cases.

Page 4, line 65: Index cases were at 73%.

Page 4, line 77: Perhaps the authors could change the heading for “overall results” to something more informative.

Page 10, line 277: Perhaps the authors could be a bit more specific when they write “to reach them”. Perhaps instead they could write “to provide services and testing for”?

Supplementary file “Link between the index case and its contact”: What does intimate mean? Does this mean a romantic relationship? Why is familial listed under intimate contact and separately as familial? Also, are the column labels missing (“initial value” and “recoded value”)?

Supplementary file “Symptoms”: what are “ear, nose, throat symptoms”? Is this a combined category that includes several of the other symptom categories? Was this a patient reported category? Or part of a close-ended checklist for participants to complete?

Page 13, line 377-378: Please clarify the following: “Infected at least one time, more than 6 months or less than 6 months. This category included persons not vaccinated but having at least one positive PCR test result, more or less than one year ago. ” Is it more than 6 or more than 12?

Methods: Perhaps the authors might consider relabeling “Building” to “multi-residential building? Apartments/condos? Also, what is a “collective structure”? Perhaps more specific labels could be used.

Reviewer #2 (Remarks to the Author):

Summary

This paper investigates the secondary attack rate (N positives / N contacts) in Switzerland June 2020 to February 2022 across 4 variants (wild type, Alpha, Delta, Omicron), which is proxied by the time period. They have 50,889 index cases (assumed to be primary cases) and 111,432 contacts (identified from contact tracing), i.e., approximately 2 contacts per index case. The authors find an increased SAR, when the inclusion period is increased (Figure S1). They find that vaccination and previous infection (i.e., natural immunity) is effective in protecting against both susceptibility to infection and infectiousness (probability of infecting others).

Major comments

1. I fail to see the overall contribution of this paper, i.e., what do we learn from it? I believe we already know that vaccinations and infections protect both against infectiousness (probability of infecting others) and against susceptibility (probability of being infected)—and the effect against susceptibility is greater than the effect against infectiousness.
2. I need some more background information. Generally, I believe the situation in Switzerland must have changed quite a lot from June 2020 to February 2022. How is the vaccination rollout? How does it correlate with age? What is the incidence rate of positive cases over time? What types of vaccinations were used? What is the test capacity over time? How do people access tests? What kind of restrictions and non-pharmaceutical interventions were at play over time?
3. The authors assume that there is only 1 dominant variant present at each time period. They

argue that new variants take over rapidly. However, they do not give any proof of this assumption. The authors define all cases after 21st December 2021 as Omicron cases. When I look up the share of Omicron cases on the internet, I find that on 20th December the Omicron share was 15%, 3rd January 61%, and 17th January 90% (cf. [https://ourworldindata.org/explorers/coronavirus-data-explorer?zoomToSelection=true&time=2020-03-01..latest&facet=none&pickerSort=asc&pickerMetric=location&Metric=Omicron+variant+\(share\)&Interval=7-day+rolling+average&Relative+to+Population=true&Color+by+test+positivity=false&country=~CHE](https://ourworldindata.org/explorers/coronavirus-data-explorer?zoomToSelection=true&time=2020-03-01..latest&facet=none&pickerSort=asc&pickerMetric=location&Metric=Omicron+variant+(share)&Interval=7-day+rolling+average&Relative+to+Population=true&Color+by+test+positivity=false&country=~CHE)). Hence, I think it is a stretch to use dates to define variants. As minimum, the authors have to use periods, where the proportion is above a certain threshold, e.g., 95%.

4. I need some more information about contacts and their compliance. How large a proportion is being tested in response to being exposed? The authors argue in the abstract (line 8) that this is important. However, I fail to find this information in the paper. Also, I find it weird that index cases have fewer contacts during the Omicron period (lines 62-64).

5. The estimates of infectiousness and susceptibility is relying on the assumption that the index case (first identified) is in fact also the primary case (first infected). The authors do not address this potential misclassification. For inspiration, see e.g., Lyngse, F.P., Mortensen, L.H., Denwood, M.J. et al. Household transmission of the SARS-CoV-2 Omicron variant in Denmark. *Nat Commun* 13, 5573 (2022). <https://doi.org/10.1038/s41467-022-33328-3>

Minor comments

1. Please proofread the paper.
2. Thousand separator should be a comma, not an apostrophe.
3. Are contacts not living outside Switzerland also included? In lines 333-338 the authors write that Geneva has a large population of international commuters (doubling the population on weekdays).
4. How do you define "previously infected" individuals? Is it conditional on being confirmed by an RT-PCR test?
5. What kind of tests are used for defining positive cases? Antigen and/or RT-PCR tests?
6. How do you classify contacts that have multiple index cases?
7. Line 136: "decreased by -9.4 percentage points". The "decreased" and "minus" cancel each other out.
8. Line 180: "transmission risk". I believe the authors mean infectiousness/contagiousness.

Minor comments

1. Please proofread the paper.
2. Thousand separator should be a comma, not an apostrophe.
3. Are contacts not living outside Switzerland also included? In lines 333-338 the authors write that Geneva has a large population of international commuters (doubling the population on weekdays).
4. How do you define "previously infected" individuals? Is it conditional on being confirmed by an RT-PCR test?
5. What kind of tests are used for defining positive cases? Antigen and/or RT-PCR tests?
6. How do you classify contacts that have multiple index cases?
7. Line 136: "decreased by -9.4 percentage points". The "decreased" and "minus" cancel each other out.
8. Line 180: "transmission risk". I believe the authors mean infectiousness/contagiousness.

Reviewer #3 (Remarks to the Author):

This study by Mongin et al. analyzed the effect of mRNA vaccination and previous infections on SARS-CoV-2 transmission across four variants leveraging unique datasets from Geneva, Switzerland. Statistical analysis was performed to disentangle the roles of the reduction of contagiousness and the increased protection against infection in reducing virus transmission, controlling for a number of variables. They found that the reduction of transmission was mainly driven by the protection of contacts against infection rather than decreased contagiousness of index cases. Natural infection conferred a larger immunity effect than vaccination and hybrid immunity. Vaccination can provide protection against infection within 6 months of administration, but only for variants before Omicron. This is an interesting and significant work, echoing many published results (as mentioned in the discussion section). However, I have a few questions for the authors to clarify and address on methods and interpretation.

1. For observational data, the key question in estimating any effect is to address observational bias. In this case, the major bias came from the differential testing among close contacts. The authors used the number of tests performed by each contact during the last 3 months preceding their contact with the index case to control for tendency to test. In total, three categories were used: 0, 1 and more than 2 (2+). Given the large variation of testing, this control seems very coarse. As the authors indicated, there are many factors affecting testing tendency. Using only three categories may not be sufficient to capture this variation. Could the authors demonstrate this control is sufficient? Are there other ways to control for testing? Is it possible to develop a

statistical model for testing propensity score based on other variables and use this score to control? Does the result robust to different control methods? My key concern is that whether observational bias was properly controlled and if the findings are robust to such control.

2. A related question. The study found that people not testing in the preceding months having a much lower SAR. Would it be possible this subpopulation is less likely to get tested so less infections were found? How to interpret this finding?

3. Another potential bias is the differential reporting of different types of contacts. Most reported contacts were from household, which usually lasted a long time. Mixing household contacts with non-household contacts (maybe less intense and shorter) in the analysis seems complicated the interpretation. Transmission risk varies a lot across settings. A discussion on this point would be helpful.

4. The analysis assumed that index cases were infectors and infected close contacts were infectees. For COVID-19, however, this is not always true. Given pre-symptomatic and asymptomatic shedding, the person first diagnosed may not be the infector. How did the authors determine the direction of transmission? I think the authors should represent such uncertainty in the estimates.

5. The Results section reads like a verbal explanation of Fig. 1 and Table 2. Maybe it's good to just highlight some major findings under each subsection. It's hard to digest so many percentage numbers.

6. In the Statistical analysis section, the authors may want to include the equation of the regression model to better communicate the method.

7. A few typos. Line 104, 4.3pp should be -4.3pp; Line 117, 5.6pp should be -5.6pp.

REVIEWER COMMENTS

Reviewer #1 (Remarks to the Author):

Summary

This was a cohort study examining factors associated with the secondary attack rate of four SARS-CoV-2 variants. The primary independent variable was immune status (immune naïve, prior infection, vaccination, hybrid immunity). Congratulations to the authors for conducting a difficult study with a large sample size.

The findings are valuable - most notably the adjusted estimates of secondary attack rates by immunity status (infection, vaccine, and hybrid immunity) for cases and contacts, demonstration of the value of the interaction term between immunity status and testing frequency (and its role in addressing confounding), and the list of variables independently associated with secondary attack rates.

These findings have relevance for understanding how to limit the future spread of SARS-CoV-2 variants and how to best time vaccinations.

The authors would like to thank the reviewer for his/her positive appreciation of our work.

I have several major and minor comments.

Major

Methods

1. Please can the authors add a summary of how long after being identified as a case the participants were contacted? (i.e., mean time to interview/form completion). This could provide insights into recall bias about their close contacts.

We added to the table 1 the mean time between the test result of the index and the first contact, which could be the telephone interview or the completion of the online form. We can actually see that this delay increases in time, thus potentially explaining part of the decrease of the number of contact declared in time.

	Overall	EU1	alpha	delta	Omicron	Missing
Number of index cases	50889	18884	6692	11628	13685	
Mean number of contact (SD) per index case	2.19 (1.90)	2.42 (2.30)	2.50 (1.83)	2.38 (1.87)	1.56 (0.93)	
Duration [days] between test result and first contact (phone or form)	1.39 (1.38)	0.77 (0.72)	1.35 (1.09)	1.68 (1.72)	2.05 (1.49)	39.1

We added to the result section:

“The index cases were contacted on average 1.4 days after their last encounter with their contacts, this delay increasing in time from 0.8 during EU wave up to 2 days during Omicron.”

We added to the discussion, limitation section:

“Another limitation comes from the changes of contact tracing during the pandemics. For example, we found an increasing delay between the index test result and the first call with advancing pandemics. Furthermore, during high epidemic activity or the Omicron wave, the indexes were not called anymore. This could modify the way the index recalled their contacts and cause potential bias.”

2. There was a lot of significance testing in the analysis. Please can the authors correct their estimates for multiple testing or justify why they have not done this correction and add this as a limitation to the discussion section?

First, we would like to emphasize that the analyses are multivariable, stratified by variant period, so the total number of models (in the main analyses) is 4. Each separate model includes 14 independent variables, corresponding to 29 degrees of freedom, for which the p value is already corrected for this number of degrees of freedom. Since in our smallest stratum, we have 3546 events, corresponding to 122 events per variable, far above any recommended events per variable. We consider these four models to apply to separate datasets since they are different variants and thus did not correct the width of the confidence intervals or the p-value threshold.

To better clarify that the models are correctly estimated in terms of the number of events per variable, we added to the discussion:

“The availability of detailed information on cases and contacts together with the high number of measures allowed adjusting for a wide range of covariates while keeping a high number of events per variable”.

In the discussion, we also added an information about type I error and number of tests.

“Our analysis considered each variant separately, to account for differences in vaccination effect across variants, which may inflate type I error due to using four models without correcting for multiple tests.”

3. Please can the authors describe whether the interview form matched the self-completion online form? If they differed, please describe the ways in which they differed and how the changes may have influenced the findings.

The interview form and the self-completion form were exactly the same. We added in the methods section:

“Additionally, an online form was implemented at the end of September 2020 to support the oral interviews, allowing the contacts to complete the required information themselves right after receiving the notification of their positive test. The interview form and the online form were identical.”

4. Did the authors have access to housing density metrics? This would be a very valuable type of

metric to include in the analysis.

We agree with the reviewer that such information would be a valuable one. We did not have access to such metric. Please note that the socio-economic indicator of the neighbourhood, which in theory should be associated with housing density, was not associated with the secondary attack rate.

Results

5. Please can the authors conduct a sensitivity analysis using the complete data (without imputation) to compare to the main results given that there was a significant amount of missing data for some variables (e.g., contact type was 38% missing)? I recognize that this is a significant amount of work so perhaps they could do this for the key analysis only to evaluate the robustness of their imputation (key analysis being the multi-variate analysis with the interaction term between contact immunity and propensity of contact to perform tests)?

We added a full « Sensitivity analysis » section in the supplementary material, which included the adjusted analysis and the interaction analysis for the complete case only. Reviewer will see that the results are similar to those of the main analysis, but with wider error bars, especially during Omicron.

We added to the methods section:

“We performed six sensitivity analysis, which can be found in the supplementary materials:

- **The first sensitivity analysis concerned imputation. We performed the same analysis but with row wise complete case data.”**

And at the end of the result section:

“Sensitivity analyses

Six sensitivity analyses were conducted to assess the influence on the results of: the imputation methods used, the definition of the variant periods, the choice of the variables to adjust for the contacts’ propensity to test, and the potential contact-case misclassification. Results of these analyses can be found in supplementary material. These analyses yielded similar results to the analyses presented above.”

Discussion

6. Please can the authors briefly describe how switching from oral interviews to self-complete interviews may have impacted the results ?

We thank the reviewer for raising this important point. During high epidemic activity, and during the omicron wave, oral interviews could not be conducted anymore. The first expected effect with the self-complete interviews is a lower number of contact declared (during oral interview, the index cases were encouraged to remember all their contacts) and a predominance of reported contacts living under the same roof, as observed in the data. A information bias could potentially alter the results if the contacts declared differ between the self-complete forms and the oral interviews in a way that is not adjusted in our analysis. We added this information bias risk in the discussion.

“The main limitation of this observational registry study is information and surveillance bias⁶².

During peak epidemics, most contact declaration were made using a self-complete online form, and less contacts were declared. Though this could be due to a real reduction in number of persons

seen during peak epidemics, it could reflect an information bias, due to the fact that self-complete forms do not elicit as many contacts as oral interviews with subsequent questions on potential contacts. We tried to account for this bias by adjusting for cases and contact characteristics.”

7. The authors may want to consider adding a limitation about selection bias. Is it possible that only moderately or mildly sick patients were able to respond to the questionnaire? If participants were very unwell (hospitalized) was it logistically possible for them to gain entry into the study? Is it possible that older people were under-represented because they were more likely to get severe disease or die and therefore not participate?

The authors would like to thank the reviewer for pointing out this potential bias. Indeed, old persons, especially those in nursing homes, and patients tested COVID+ directly in the hospital were less likely to be contacted by the COVID-19 contact tracing unit because these persons were handled locally by their respective health institution.

We added in the limitation:

“As previously mentioned, people over 65 years old are underrepresented, while young people are overrepresented. The underrepresentation of old people may be due to the handling of contact tracing and isolation by their specific nursing home or healthcare facility, or due to their health status which did not allow them to provide their contacts. As a consequence, vaccinated people, who tend to be older, were also underrepresented in our cohort. This could potentially lead to selection bias, although we tried to adjust for most of the factors potentially influencing the SAR.”

8. Was there any other severity information available in the registry? If all the participants that were very unwell or that died were excluded from the study, then it’s possible that these results only apply to persons with mild or moderate disease.

We added to the table 1 the number of index person who died from COVID-19. The reviewer can see that 206 index cases had declared contacts and died from COVID-19 (death from COVID-19 certified by a physician), corresponding to 0.4% of the index. This percentage corresponds to the percentage of deaths in the total register at the end of 2022 (roughly 950 COVID-19 deaths for 250000 cases, that is 0.4%), indicating that the subset of index declaring contacts does not appear to bias towards lower severity of the disease.

	Overall	EU1	Alpha	Delta	Omicron	Missing
Number of index cases	50889	18884	6692	11628	13685	
Index who died from COVID-19	206 (0.4%)	49 (0.6%)	27 (0.2%)	122 (0.7%)	8 (0.1%)	0
Total number of contacts	111432	45755	16755	27636	21286	
Number of contact who died from COVID-19	130 (0.1%)	20 (0.1%)	18 (0.1%)	86 (0.2%)	6 (0.03%)	0

We added these reassuring results (in terms of study validity and selection bias) in the limitation section.

“Though there could be selection bias, the case fatality rate of the index who reported contacts (0.4%) corresponds to the overall case fatality rate in the register, indicating that the severity of the cases reporting contacts seem similar to the overall COVID19 positive population.”

Minor

Page 3, Line 35-36: Consider deleting “Western countries” or changing it to “high income countries”. Boosters have been implemented for the purpose of maintaining immunity regardless of where they are being administered.

Done

Page 4, line 62: The resulting dataset consisted of 50’889 index cases.

Done

Page 4, line 65: Index cases were at 73%.

The sentence now reads:

“Index cases comprised 73% adults between the ages of 18 and 64 ...”

Page 4, line 77: Perhaps the authors could change the heading for “overall results” to something more informative.

We modified “overall results” to “secondary attack rate”

Page 10, line 277: Perhaps the authors could be a bit more specific when they write “to reach them”. Perhaps instead they could write “to provide services and testing for”?

We modified the sentence as follow:

“Finally, the canton of Geneva invested a lot of effort in testing and following vulnerable populations during the pandemics, including undocumented migrants, thereby reducing potential selection bias.”

Supplementary file “Link between the index case and its contact”: What does intimate mean? Does this mean a romantic relationship? Why is familial listed under intimate contact and separately as familial? Also, are the column labels missing (“initial value” and “recoded value”)?

Yes intimate means a romantic relationship. Thank you for pointing out these mistakes, indeed Familial relation is under the new category intimate or familial. The table is now:

Initial value	Recoded value
Live under the same roof	Same roof
Intimate contact	Intimate or familial
Familial relation	Intimate or familial
Professional relation	Other
School	Other
University	Other
Leisure/hobbies	Other
Healthcare	Other
Other	Other

Supplementary file “Symptoms”: what are “ear, nose, throat symptoms”? Is this a combined category that includes several of the other symptom categories? Was this a patient reported category? Or part of a close-ended checklist for participants to complete?

Ear nose throat symptoms is a general variable that was applied at the beginning of the pandemic, when there was no precise idea of the COVID-19 symptoms. Thereafter, runny/stuffy nose, sore throat, and loss of taste/smell symptoms were better characterized and these categories were added in the list of symptoms. From May 2020, ear, nose throat symptoms was no longer used and was replaced by the above-mentioned detailed variables. We added this information in supplementary material.

Page 13, line 377-378: Please clarify the following: “Infected at least one time, more than 6 months or less than 6 months. This category included persons not vaccinated but having at least one positive PCR test result, more or less than one year ago. “ Is it more than 6 or more than 12?

It is 6, sorry for the mistake, we corrected it.

Methods: Perhaps the authors might consider relabeling “Building” to “multi-residential building? Apartments/condos? Also, what is a “collective structure”? Perhaps more specific labels could be used.

We reformulated our description as follow:

“The type of building was categorised in three categories: buildings (multi-residential buildings, potentially having shops), single houses, or collective structures. This last category included nursing homes, jails, asylums and fire-stations.”

Reviewer #2 (Remarks to the Author):

Summary

This paper investigates the secondary attack rate (N positives / N contacts) in Switzerland June 2020 to February 2022 across 4 variants (wild type, Alpha, Delta, Omicron), which is proxied by the time period. They have 50,889 index cases (assumed to be primary cases) and 111,432 contacts (identified from contact tracing), i.e., approximately 2 contacts per index case. The authors find an increased SAR, when the inclusion period is increased (Figure S1). They find that vaccination and previous infection (i.e., natural immunity) is effective in protecting against both susceptibility to infection and infectiousness (probability of infecting others).

Major comments

1. I fail to see the overall contribution of this paper, i.e., what do we learn from it? I believe we already know that vaccinations and infections protect both against infectiousness (probability of infecting others) and against susceptibility (probability of being infected) and the effect against susceptibility is greater than the effect against infectiousness.

The authors agree with the reviewer that the protection against susceptibility conferred by vaccine and previous infection and its waning in time has been established for several VoCs in several recent publications that we cite in our article.

We respectfully disagree with the reviewer about the protection against infectiousness:

- First there are mixed results in the literature, with various large studies presenting no significant effect of vaccination on infectiousness (see [https://doi.org/10.1016/S1473-3099\(21\)00648-4](https://doi.org/10.1016/S1473-3099(21)00648-4) , <https://doi.org/10.1126/science.abl4292> or <https://doi.org/10.1001/jamanetworkopen.2022.9317>)
- Some other studies report an effect of vaccination on infectiousness, such as this early study <https://doi.org/10.1056/NEJMc2107717> or the more recent reference that the reviewer also indicated <https://doi.org/10.1038/s41467-022-33328-3>. But these studies do not consider previous infections, nor the potential waning of immunity, nor compare the effect on infectiousness with the effect on susceptibility. Comparison between the effect of vaccination and infection- on infectiousness, and their evolution in time, has been suggested in a study about viral load (<https://doi.org/10.1038/s41467-022-33096-0>), but to our knowledge not observed in epidemiological data

Our study is one of the first to (clearly) demonstrate that immunity conferred by vaccination and previous infection reduce infectiousness and that this effect is lower than the reduction of susceptibility but less sensitive to time and to VoC changes.

Second, contrary to the reviewer's suggestion, we actually show that the ratio of the effect of vaccination against susceptibility and infectiousness change along the variant and that it is reversed for Omicron (effect against infectiousness is higher than against susceptibility).

Last, our study adjusts for a large panel of potential confounders, such as the testing of the contacts, which is an important source of bias that is not accounted for in the majority of the studies. We thus think that our study is novel in the sense that it offers the first complete picture of the effects of both vaccination and previous infection on infectiousness and susceptibility and its evolution in time, across 4 VoCs while adjusting for the main confounders.

This being said, we agree with the reviewer that we could better explain the novelty of our study. To insist more on these points, we modified the introduction as follows:

“Less is known, however, on the effect of immunity on the probability to contaminate others (infectiousness), especially with regard to natural immunity¹⁹. Depending on the variant of concern (VoC) considered, studies analysing the secondary attack rate show contrasting results, from no effect of vaccination on infectiousness^{20–22} to a clear reduction in the attack rate²³. The effect of previous infection on the reduction of infectiousness and its evolution over time is unclear, while recent in vitro studies measuring viral load and propagation indirectly suggest that natural infection could reduce infectiousness better than vaccination^{24,25}. Similarly, little is known about how the reduction of infectiousness and the reduction of susceptibility conferred by the immunity compare in the reduction of SarS-CoV-2 transmission.”

We also added a paragraph in the result section along with a new figure (figure 2, reproduced below):

“Of note, the effect of immunity on SARS-CoV-2 propagation is shared with a 1:3 ratio between the reduction of infectiousness and the reduction of infection susceptibility (see Figure 2) for previous infections, but not for recent vaccination. Indeed, this ratio seems to decrease with new variants, and is even reversed for Omicron, where recent vaccination no longer has an effect on susceptibility but an increased effect on infectiousness.”

And modified the discussion as follow:

“The main immune factor lowering the secondary attack rate was natural infection, while vaccination had a more limited impact, even when recent enough. The reduction of infectiousness conferred by vaccination appears to wane less in time and to be less sensitive to variant changes than the decrease of infection susceptibility, making this effect the major contribution of vaccination to the reduction of Sars-CoV-2 propagation for Omicron.”

2. I need some more background information. Generally, I believe the situation in Switzerland must have changed quite a lot from June 2020 to February 2022. How is the vaccination rollout? How does it correlate with age? What is the incidence rate of positive cases over time? What types of vaccinations were used? What is the test capacity over time? How do people access tests? What kind of restrictions and non-pharmaceutical interventions were at play over time?

The authors agree with the reviewer that more general information should be provided. We provide, in supplementary material, the evolution during the period considering of the number of cases, number of tests and number of deaths from COVID-19 in Geneva (see graph below). We also briefly described the vaccination strategy and roll-out, as well as the main NPIs implemented. We hope that these information will allow a better understanding of our study and will meet the reviewer's needs.

3. The authors assume that there is only 1 dominant variant present at each time period. They argue that new variants take over rapidly. However, they do not give any proof of this assumption. The authors define all cases after 21st December 2021 as Omicron cases. When I look up the share of Omicron cases on the internet, I find that on 20th December the Omicron share was 15%, 3rd January 61%, and 17th January 90% (cf. [https://ourworldindata.org/explorers/coronavirus-data-explorer?zoomToSelection=true&time=2020-03-01..latest&facet=none&pickerSort=asc&pickerMetric=location&Metric=Omicron+variant+\(share\)&Interval=7-day+rolling+average&Relative+to+Population=true&Color+by+test+positivity=false&country=~CHE](https://ourworldindata.org/explorers/coronavirus-data-explorer?zoomToSelection=true&time=2020-03-01..latest&facet=none&pickerSort=asc&pickerMetric=location&Metric=Omicron+variant+(share)&Interval=7-day+rolling+average&Relative+to+Population=true&Color+by+test+positivity=false&country=~CHE)). Hence, I think it is a stretch to use dates to define variants. As minimum, the authors have to use periods, where the proportion is above a certain threshold, e.g., 95%.

The authors agree with the reviewer that the manuscript did not provide enough information about the definition of the variant periods.

As specified in the manuscript, the underlying data stem from Global Initiative on Sharing All Influenza Data (GISAID, <https://gisaid.org/>) and are provided by the website <https://covariants.org/per-country?region=Switzerland>. These data differ from the one from ourworldindata.org, because in covariant.org, Switzerland is divided in 6 regions, whereas the data the reviewer is referring to provide an average situation over the whole Switzerland.

The data we use from covariant/GISEAD concern the regions of Geneva, Vaud and Valais, and stem mainly from wastewater analysis. In the previous version of the manuscript, we used the online graphic to determine approximately the date when a VoC became dominant (i.e. above 50%). We propose in the present revision a more solid and reproducible definition of the variant period. The data concerned are extracted from https://github.com/hodcroftlab/covariants/blob/master/cluster_tables/SwissClusters_data.json (the underlying data of covariant.org). After selecting the data for the region1 of Switzerland and the four variant of interest in our study, we modelled the VoC evolution as a rising and decreasing sigmoid in time. This approach yields the estimations presented in the graph below (estimations are solid lines, points are the experimental data)

We then used the estimation curves to define the boundary dates for two different levels of predominance.

In the main analysis, we kept a level of 50%. The dates differs slightly from previous version, and we updated our analysis with the new dates:

VoC	EU	alpha	delta	Omicron
Start	01-06-2020	06-01-2021	15-06-2021	18-12-2021
end	05-01-2021	14-06-2021	17-12-2021	01-02-2022

We then performed a sensitivity analysis, with periods defined for >90% dominance (see figure above). In this case, the dates are

VoC	EU	alpha	delta	Omicron
Start	01-06-2020	01-02-2021	01-07-2021	31-12-2021
end	04-12-2020	19-05-2021	01-12-2021	01-02-2022

The results of this sensitivity analysis are presented in detail in supplemental material and are not different from the main analysis.

The code used to perform the regression of the variant evolution in time can be found with the analysis code in the GitLab repository, together with the underlying GISEAD data.

We modified the method section as follow:

“As the ARGOS data did not contain information about the SARS -CoV-2 variant type, we divided the study periods according to the predominance of the SARS-CoV-2 variant of interest, based on the data provided by the Global Initiative on Sharing All Influenza Data⁸⁰ for the Geneva region, which stem mainly from wastewater analysis. To do this, we modelled the evolution of the share of variants as rising and falling sigmoids (see supplementary figure S1), and determined the period of predominance with one VoC above 50%.

- EU1 from 01-06-2020 to 05-01-2021
- Alpha from 06-01-2021 to 14-06-2021
- Delta from 15-06-2021 to 17-12-2021
- Omicron from 18-12-2021 to 01-03-2022 (mainly BA.1)

In a sensitivity analysis we considered also periods defined with a threshold of 90% (see sensitivity analysis section).”

4. I need some more information about contacts and their compliance. How large a proportion is being tested in response to being exposed? The authors argue in the abstract (line 8) that this is important. However, I fail to find this information in the paper. Also, I find it weird that index cases have fewer contacts during the Omicron period (lines 62-64).

The reviewer is right, thank you for noticing. We added to the table 1 the number of contacts who performed a test during the 10 days after being exposed.

	Overall	EU1	Alpha	Delta	Omicron	Missing
Number of contacts who performed a test during the 10 days following their contact with the index	46417 (41.6%)	13392 (31.7)	12808 (63.1)	11043 (40.5)	9174 (42.1)	0.0

We modified the result section as follow:

“Among the 111’432 declared contacts, 46’417 performed a test during the 10 days following the date of the last contact with the index case and 21’387 had a positive test result (raw SAR of 19.2%). This raw SAR increased almost linearly of 3 percent point per day when increasing the delay from 0 to 8 days, to then plateauing after 10 days (see supplementary figure S1). For the rest of the study, a delay of 10 days was considered. The raw SAR changed across variants and was 16.5% during the EU1 wave, 21.2% during the Alpha wave, 16.8% during the Delta wave, and 26.3% during the Omicron wave. Of note, the proportion of contacts performing a test during the period of interest evolved over time and was 31.7% during EU1, 63.1% during Alpha, 40.5% during the Delta wave and 42.1% during the Omicron wave.”

The decrease in time of the number of contact declared is expected and has been reported in various studies, and has different origins:

- First, citizens lose their desire to participate to restrictive measures that last in time, thus the slow decrease in time of the number of contacts reported.
- Vaccination can also play a role: vaccinated people did not have to perform quarantine anymore and were considered protected from reinfection at the time, which is something that surely lowered the number of declared contacts
- Last, as reported in the manuscript, during omicron the index cases could not be called by the COVID-19 unit anymore because of the high number of positive cases to handle. Declaration of contacts relied only on auto-completion forms, which are known to be less efficient in term of number of contacts reported.

For a similar remark of reviewer 1, we added the following paragraph to the limitation:

“Another limitation comes from the changes of contact tracing during the pandemics. For example, we found an increasing delay between the index test result and the first call with advancing pandemics. Furthermore, during high epidemic activity or the Omicron wave, the indexes were not called anymore. This could modify the way the index recalled their contacts and cause potential bias.”

5. The estimates of infectiousness and susceptibility is relying on the assumption that the index case (first identified) is in fact also the primary case (first infected). The authors do not address this potential misclassification. For inspiration, see e.g., Lyngse, F.P., Mortensen, L.H., Denwood, M.J. et al. Household transmission of the SARS-CoV-2 Omicron variant in Denmark. Nat Commun 13, 5573 (2022). <https://doi.org/10.1038/s41467-022-33328-3>

The authors would like to thank the reviewer for this pertinent remark.

In a similar approach to the reference indicated, we performed three sensitivity analyses to address three potential misclassifications:

- To address misclassification of community cases, we restricted our analysis to contact cases that were placed in quarantined at maximum the day after their last contact with the index case
- To address potential misclassification of primary case, we performed a sensitivity analysis with a more restrictive definition of the contact being positive, considering that the contact become positive if they have a positive result at least 4 days after its last contact with the index case (and less than 10 days after)
- To address potential misclassification of tertiary case, we restricted our analysis to household with only one contact

These three sensitivity analysis are described in the method section, and their detailed results can be found in supplementary material. Their results do not differ from the main analysis, confirming the results of the main analysis.

Minor comments

1. Please proofread the paper.

We did our best to proofread the paper.

2. Thousand separator should be a comma, not an apostrophe.

Done

3. Are contacts not living outside Switzerland also included? In lines 333-338 the authors write that Geneva has a large population of international commuters (doubling the population on weekdays).

Contacts living outside Switzerland are not included. The reason is that they are likely to be tested in France. As our register includes all tests performed in Geneva, including contacts not living in Geneva would bias the SAR towards lower values.

4. How do you define “previously infected” individuals? Is it conditional on being confirmed by an RT-PCR test?

Yes, previously infected is conditioned on having a positive test, which is in the vast majority PCR test.

5. What kind of tests are used for defining positive cases? Antigen and/or RT-PCR tests?

We considered both. The vast majority of the tests are PCR tests, and most of the antigen tests are subsequently confirmed by PCR tests in the majority of the cases.

For the positive contacts, 91% of the results are PCR. For the index cases, the same proportion are found: 91% of PCR.

6. How do you classify contacts that have multiple index cases?

We consider in our analysis an index in a contact dyad. Therefore, a contact can appear in various dyads, and so can the index.

We added in the “methods” section, “Index cases and contacts” subsection:

“In the present study, the dataset is composed of the index case and contact dyads residing in Geneva. An index case could appear for various infections, and a contact could appear with various index cases.”

7. Line 136: “decreased by -9.4 percentage points”. The “decreased” and “minus” cancel each other out.

Thank you, we deleted the “minus” in front each time there was a “decrease” or “lower” before.

8. Line 180: “transmission risk”. I believe the authors mean infectiousness/contagiousness.

Yes the reviewer is right. We now use more systematically, as suggested by the reviewer in his/her introduction paragraph, the terms infectiousness/contagiousness and infection susceptibility.

Reviewer #3 (Remarks to the Author):

This study by Mongin et al. analyzed the effect of mRNA vaccination and previous infections on SARS-CoV-2 transmission across four variants leveraging unique datasets from Geneva, Switzerland. Statistical analysis was performed to disentangle the roles of the reduction of contagiousness and the increased protection against infection in reducing virus transmission, controlling for a number of variables. They found that the reduction of transmission was mainly driven by the protection of contacts against infection rather than decreased contagiousness of index cases. Natural infection conferred a larger immunity effect than vaccination and hybrid immunity. Vaccination can provide protection against infection within 6 months of administration, but only for variants before Omicron. This is an interesting and significant work, echoing many published results (as mentioned in the discussion section). However, I have a few questions for the authors to clarify and address on methods and interpretation.

The authors would like to thank the reviewer for his/her positive appreciation of our work

1. For observational data, the key question in estimating any effect is to address observational bias. In this case, the major bias came from the differential testing among close contacts. The authors used the number of tests performed by each contact during the last 3 months preceding their contact with the index case to control for tendency to test. In total, three categories were used: 0, 1 and more than 2 (2+). Given the large variation of testing, this control seems very coarse. As the authors indicated, there are many factors affecting testing tendency. Using only three categories may not be sufficient to capture this variation. Could the authors demonstrate this control is sufficient? Are there other ways to control for testing? Is it possible to develop a statistical model for testing propensity score based on other variables and use this score to control? Does the result robust to different control methods? My key concern is that whether observational bias was properly controlled and if the findings are robust to such control.

The reviewer raises an important point. We did not find many other proper way to adjust for the testing. We cannot adjust for performing a test after being exposed, as this would be directly linked to the outcome. The authors do not consider that a propensity score would be a good option here, because our multivariable model already adjust for many variables that are likely to become correlated to the score we may want to calculate.

However, in order to answer the reviewer suggestions, the authors propose a sensitivity analysis, where we use the number of tests performed the last 6 months instead of the last 3 that we categorize in 4 categories: 0, 1, 2 and 3+.

The descriptive statistics of this new variable are (and are presented in supplementary material):

	Overall	EU1	Alpha	Delta	Omicron	Missing
Number of tests performed the last 6 months (%)						0
1	24680 (22.1)	5008 (24.7)	6171 (22.6)	8244 (19.5)	5257 (24.1)	
0	73097 (65.5)	12689 (62.5)	16630 (61.0)	32202 (76.1)	11576 (53.1)	
2	8511 (7.6)	1745 (8.6)	2578 (9.5)	1490 (3.5)	2698 (12.4)	
3+	5386 (4.8)	869 (4.3)	1881 (6.9)	359 (0.8)	2277 (10.4)	

The detailed results of this new multivariable analysis, with and without interaction term between testing and immunity, can be found in supplementary material. The analysis yield results very similar to the main analysis, comforting the robustness of the presented results.

2. A related question. The study found that people not testing in the preceding months having a much lower SAR. Would it be possible this subpopulation is less likely to get tested so less infections were found? How to interpret this finding?

Yes exactly. To be more precise, our interaction study evidences that this is true among non-immune population: we see a clear lower SAR among the populations that test less, because less infections are found (the probability that they test after being exposed is lower).

This is not the case among immune contacts (infected or vaccinated): the SAR is similar between populations testing the preceding months or not, showing that the protection conferred by immunity is stronger than the effect of the tendency to get tested.

We explained the above in more detail, and modified the discussion as follow:

“The main confounder of this association was the tendency of the contact to test, which modified the SAR of the non-immune population, our reference category. Indeed, the SAR was much higher among non-immune people who tested compared with those who did not, because they were more likely to test after being exposed to the index case. But the SAR was quite similar among previously infected or recently vaccinated people, irrespective of their tendency to test, suggesting that the protective effect of immunity was stronger than the tendency to test.”

3. Another potential bias is the differential reporting of different types of contacts. Most reported contacts were from household, which usually lasted a long time. Mixing household contacts with non-household contacts (maybe less intense and shorter) in the analysis seems complicated the interpretation. Transmission risk varies a lot across settings. A discussion on this point would be helpful.

We absolutely agree with the reviewer, and this is why we adjusted for the contact types. This allows to determine common influencing factor across contact types while differentiating them. Our analysis yields a clear difference in SAR between the types of contact, where the SAR is much higher for household contacts than for the others.

We realised that our introduction did not clearly state that we adjusted for the types of contact, nor did the method section give the proper list of all control variables. We deeply thank the reviewer for his/her careful reading of our work.

We modified the introduction as follow:

“[...]we propose to study the effect of the immune status on SARS-CoV2 secondary attack rate (SAR) along 4 SARS-CoV-2 variants, considering vaccination and natural infection of index and contacts while adjusting for demographic, social, health factors as well as the contact settings and the tendency to test for SARS-CoV-2.”

The control subsection of the methods section now reads:

“The estimation of the effect of immunity of both contact and index on SAR was controlled by the age and gender of the index and contact, the body mass index (BMI) of the index, the presence of

symptoms and cough for the index, the type of building in which the index is living, the neighbourhood socio-economic condition of the index, personal vulnerability of the index, the type of relation between index and contact, and the propensity of the contact to test.

Age was categorised in three categories: 0-17 years, 18-64 years and above 65 years (65+).

BMI was calculated from height and weight and was categorized in obese and non-obese categories. For age superior to 18 years, obese was considered for BMI above 30 kg/m². For age below 18, we used the extended international body mass index cut-offs corresponding to the threshold of 30 kg/m² at 18 years old ⁷⁹.

Presence or absence of symptoms was operationalized as 1 if the person reported any symptoms, otherwise 0. Cough was defined as the presence of dry or wet cough symptoms.

Categorization of the socio-economic condition of the neighbourhood area (417 official neighbourhood areas in the State of Geneva) was, similarly to previous work ⁵³, based on an index provided by the centre for the analysis of territorial inequalities (see supplementary material). The statistical office of Geneva provided the type of building and number of inhabitant for each address. The building type were categorised in three categories: building (multi-residential building, potentially having shops), single houses, or collective structure. This last category included nursing homes, jails, asylums and fire-stations.

A person was considered vulnerable is the person reported difficulty to make ends meet, lived in a highly subsidized housing, or if they asked explicitly to avoid police control.

The type of relationship between index and contact was implemented in three categories: living under the same roof, having an intimate or familial relationship, or other relationship. The correspondence between the initial categories available in the dataset and the three categories for the present study is described in the supplementary material.

Tendency to test was estimated by counting the number of tests performed by each contact during the last 3 months preceding their contact with the index case. This number was categorized in three categories: 0, 1 and more than 2 (2+). In a sensitivity analysis, we also considered the number of tests performed in the last 6 months, categorized into four categories: 0, 1, 2 and more than 3 (3+), see sensitivity analysis subsection."

The matter was already discussed in the introduction:

"Apart from the immunity of the population and the VoC considered, SAR is known to vary greatly by contact settings, ranging from 20% in households to 6% in social gatherings during the first year of the pandemic ²⁶⁻²⁹, [...]"

In our discussion, we added the following:

"The context of the encounter between the index and the contacts greatly affected the SAR, with more distant relationships (work, leisure) leading to lower SAR than housing relation, as noticed elsewhere ²⁹. The policy implication of this finding could be that requiring a quarantine only of household contacts instead of all contacts is an appropriate solution to reduce the burden of health policy without increasing significantly the transmission. "

4. The analysis assumed that index cases were infectors and infected close contacts were infectees. For COVID-19, however, this is not always true. Given pre-symptomatic and asymptomatic shedding, the person first diagnosed may not be the infector. How did the authors determine the direction of transmission? I think the authors should represent such uncertainty in the estimates.

The authors fully agree with the reviewer' comment. Reviewer 2 has a similar comment (see point 5 of reviewer 2). To address potential misclassification we performed 3 additional sensitivity analysis:

- To address misclassification of community cases, we restricted our analysis to contact cases that were placed in quarantine at maximum the day after their last contact with the index case
- To address potential misclassification of primary case, we performed a sensitivity analysis with a more restrictive definition of the contact being positive, considering that the contact become positive if they have a positive result at least 4 days after its last contact with the index case (and less than 10 days after)
- To address potential misclassification of tertiary case, we restricted our analysis to household with only one contact

These three sensitivity analysis are described in the method section, and their detailed results can be found in supplementary material. Their result do not differ from the main analysis, confirming our approach.

5. The Results section reads like a verbal explanation of Fig. 1 and Table 2. Maybe it's good to just highlight some major findings under each subsection. It's hard to digest so many percentage numbers.

Thank you. We tried to simplify the results, by merging various section and describing less in detail non-essential results.

6. In the Statistical analysis section, the authors may want to include the equation of the regression model to better communicate the method.

We added to the statistical section:

“The independent variables of the regression were the immune status of the index case and its contact, and the control variables were the age and gender of the index and contact, the body mass index (BMI) of the index, the presence of symptoms and cough for the index, the type of building in which the index lives, the neighbourhood socio-economic condition of the index, personal vulnerability of the index, the type of relationship between index and contact, and the propensity of the contact to test.”

7. A few typos. Line 104, 4.3pp should be -4.3pp; Line 117, 5.6pp should be -5.6pp.

Thank you again for your careful reading, we corrected it.

REVIEWER COMMENTS

Reviewer #1 (Remarks to the Author):

Thank you for inviting me to review a revised version of the manuscript. The authors did an excellent job responding to my comments.

Reviewer #2 (Remarks to the Author):

Major comments

1. I am not convinced by the author's causal inference framework. How can we be sure that 1) the index case is in fact the primary case, 2) the never-tested contacts are in fact not infected, 3) the positive contacts are in fact secondary cases to the index case and not infected by a third party? Line 388-389 states that "An index case could appear for various infections, and a contact could appear with various index cases", i.e., the same contact can have several primary cases.

2. Obtaining a test is a necessary condition for testing positive. Hence, understanding the testing dynamics of index cases and contacts are essential. E.g., I would like to see the testing dynamic, i.e., when do contacts get tested relative to being contact traced. There should be little to no pre-trend in the days prior to contact tracing and then sharp increases in the days after. Moreover, I would like to see the overall testing propensity of contacts by controls, e.g., age of index case, age of contact, and over time.

Index cases report fewer than two contacts on average. I find this number quite low, as it includes all contacts within the preceding 10 days, e.g., household members, work colleagues, school classmates, etc. Furthermore, conditional on being contact traced, less than 50% of the contacts are tested. This links to line 287-289, where the authors state that Geneva has invested a lot in testing.

These analyses affect the selection bias of both index cases, contacts and tested contacts.

3. I believe that time is an important factor here. E.g., for Delta, the period spans July through November. I would believe the contact patterns are different, as there must be differences in outdoor activities, school holidays, etc. The authors also write that contact tracing effort and test capacity change over time. These temporal changes should be addressed.

Did the definition of contacts change over time or due to vaccination status? Did testing guidelines change over time or due to vaccination status? How many tests were a contact recommended to obtain after being contact traced?

4. I believe the authors should use the "variant definition period" with >90% dominance cut-off. Using the >50% dominance cut-off allows for too much misclassification across variants. On the other hand, if the author could obtain whole-genome sequence data on the RT-PCR test level, the overlapping periods would allow them to validate their causal model.

5. The authors define natural immunity as individuals that have a previous positive RT-PCR test. However, there must have been many infected individuals from the first wave that never obtained a positive PCR test. These individuals are then 'wrongly' classified as immune-naïve until they become vaccinated or become infected again (identified with a PCR test).

6. I am not sure how the authors can argue that the effect of vaccinations is larger on infectiousness compared to susceptibility. The measure for infectiousness is conditional of a breakthrough infection. Thus, the group of individuals used for the susceptibility estimates is different from the group of individuals used for the infectiousness estimates.

Minor comments

1. The paper needs to be proofread.

SARS-CoV-2 is spelled in different ways throughout the paper, e.g., Sars-CoV-2, SARS-Cov-2, SARS-CoV-2, SarS-CoV-2, SARS-CoV2. In the abstract, it is spelled in two different ways!

Line 25, burden should be plural.

Line 220, delta and omicron should be capitalized.

2. Please provide the estimation equation.

3. I find the age groups to be pretty coarse. How similar are 20 year olds and 60 year olds with respect to access to vaccines, contact patterns (e.g., having small children), health status, etc.?

4. How are partially vaccinated individuals grouped, e.g., an individuals with first dose but not

- second dose? Why are booster-vaccinated individuals grouped together with other vaccinated individuals?
5. The authors use several different definitions that might or might not refer to the same thing. What is the difference between infectiousness, contagiousness, infectivity, and transmissibility? What is the difference between being contaminated and being infected?
 6. Figure S1. What does "experimental" data mean?
 7. Figure S2 should probably be in fractions of the population.
 8. Figure S3. Did Geneva not have any school closures?
 9. What is the hierarchy of contacts, e.g., is an intimate partner living in the same house classified as "household" or "intimate"?
 10. The paper concludes that we need to install ventilation or air filtration and that it is effortless to install. I find no basis for the effect of these measurements nor that it is effortless/free to install in all buildings in Geneva.
 11. I do not understand this sentence "A coughing index increased the SAR by 4.9pp [4.0, 5.8] and 6.4pp [5.0, 7.9] for the EU1 and alpha variant, but this effect was reduced to 2.3pp [1.2, 3.5] for the delta variant and 1.6pp [0.1, 3.0] for Contact children had a lower SAR, especially for early variant." (line 167-170)
 12. Line 178. Reference to Figure S3 should be Figure S6.
 13. It is very difficult to compare the sensitivity analyses in the appendix to the analyses in the main manuscript.
 14. The supplementary appendix is formatted in A3, not in A4.
 15. The title of the supplementary appendix uses an apostrophe and not a comma as thousand separator.
 16. I would like more detailed information on the vaccine rollout, including dates (e.g., months) and groups of individuals (e.g., 80+ year, 70+ year). In addition, I failed to find any information of the take-up rate.
 17. Line 92. I do not understand what "two asymptomatic adult men" refers to. Is it both the contact and index case?
 18. I am not sure how to think about nursing homes, jails, etc. I would think the contact patterns and access to tests and isolation is very different in these kind of settings.
 19. What is the time difference between test and test result?
 20. The type of person that has not been infected or vaccinated is very different over time.
 21. It seems weird that almost the same proportion of contacts were immune-naïve (NVNI) during Delta and Omicron (53% vs. 54%), cf. Table 1.
 22. Please briefly state the types of vaccines used in the main manuscript, including the take-up rate. At the moment, the reader has to go to the supplementary appendix.

Reviewer #3 (Remarks to the Author):

The authors have addressed my questions. I appreciate their efforts.

Major comments

1. I am not convinced by the author's causal inference framework. How can we be sure that 1) the index case is in fact the primary case, 2) the never-tested contacts are in fact not infected, 3) the positive contacts are in fact secondary cases to the index case and not infected by a third party? Line 388-389 states that "An index case could appear for various infections, and a contact could appear with various index cases", i.e., the same contact can have several primary cases.

Answer: We agree with the reviewer on all 3 points, there is measurement error and uncertainty on all SARS-Cov2 tests, and on the order of infection. Unfortunately, these issues cannot be entirely solved, though the reviewer's suggestion in the last round to follow the procedure of <https://doi.org/10.1038/s41467-022-33328-3> led us do perform three sensitivity analysis which provided evidence on the robustness of our results against the main potential misclassifications. The line 388-389 was added after the last review round to answer reviewer 1 comments and was meant to explain the structure of the data, where a line is an index/contact relation. As the study spans 2 years of pandemics, a contact person can be associated with several index cases during these 2 years. We found little cases where a contact was declared by two index cases infected few days apart (among the 111674 contacts, 4800 persons were declared by 2 or more index less than 2 days apart).

We emphasized in the results the limitation in terms of measurement error, line 336:

A third source of information bias is the supposition that the index cases are the primary cases, and that the contacts becoming positive in less than 10 days after their last contact with the index cases are the secondary cases. Though we cannot rule out misclassification (some contacts may actually be the primary cases or tertiary case), the three sensitivity analyses performed to address potential misclassifications indicate that our results are robust to this measurement error.

2. Obtaining a test is a necessary condition for testing positive. Hence, understanding the testing dynamics of index cases and contacts are essential. E.g., I would like to see the testing dynamic, i.e., when do contacts get tested relative to being contact traced. There should be little to no pre-trend in the days prior to contact tracing and then sharp increases in the days after. Moreover, I would like to see the overall testing propensity of contacts by controls, e.g., age of index case, age of contact, and over time.

Index cases report fewer than two contacts on average. I find this number quite low, as it includes all contacts within the preceding 10 days, e.g., household members, work colleagues, school classmates, etc. Furthermore, conditional on being contact traced, less than 50% of the contacts are tested. This links to line 287-289, where the authors state that Geneva has invested a lot in testing. These analyses affect the selection bias of both index cases, contacts and tested contacts.

Answer: The authors agree that obtaining a test is a central point for the outcome considered. The authors would like to emphasize that, aware of the matter, their study is one of the few to account for the propensity of the contact to test when estimating the SAR and dedicated a significant part of the analysis to the matter.

The authors would also like to point out that to answer a previous comment of the reviewer, they already provided the number of contacts who performed a test during the 10 days following the encounter with their index case, while the number of tests during the 3 months prior of being a contact was provided in the initial draft.

We here provide the testing proportion by age of index case, age of contact and over time.

	Did not test during the 10 days	Tested during the 10 days
N	65257	46417
Contact age (%)		
18-64	32131 (58.7)	30978 (66.8)
0-17	19675 (35.9)	13305 (28.7)
65+	2943 (5.4)	2110 (4.5)
Index age (%)		
18-64	48567 (74.4)	32777 (70.6)
0-17	13712 (21.0)	11627 (25.1)
65+	2975 (4.6)	2011 (4.3)

Concerning the age of those testing, as expected, children tend to test less (they did not need to quarantine and were considered less at risk). This result is already discussed in the discussion, as it is consistent with the effects. Concerning the index age, there is not much difference. The important peak at day 7 stems from the fact that from 8th February, 2021 to 31st of December 2021, the quarantine could be shortened with a negative test a day 7 (see our answer to your point 3 and information associated). All this information has been added to supplementary materials.

We added the graph and the table to the supplementary materials, and referred to the graph above in the results:

The number of tests performed by the contacts increased strongly during a period starting one day before the last contact with the index and decreasing back 10 days after (supplementary figure S6).

With respect to the number of contacts reported per index:

- as stated in the first paragraph of the results and in table 1, on average, index report more than 2 contacts during the whole duration (and not less): “The mean number of declared contact per infected person was 2.2 overall, with a net decrease during the Omicron period (1.6 mean contacts per index, see table 1).”
- our study is on the upper range of number of contacts declared for similar study. Looking at the numbers reported in the literature, as those provided for example by Madewell and co-authors in their systematic review on SAR

(<https://jamanetwork.com/journals/jamanetworkopen/fullarticle/2791601> , etable 2 of supplementary material), the reviewer will see that :

- *There is only one study over the 135 reported with higher number of index cases than ours (De Gier et al)*
- *Almost all studies with thousands of index cases (so with comparable real world data) report around 2 contact per index or less: 142,540 contacts for 85210 index for De Giers¹¹, 7130 contacts for 4921 index for de Gier¹², 23156 contacts for 11939 index for Lyngse²⁹, 17945 contacts for 8541 index for Lyngs³¹, 1299 contacts for 1122 index Jalali²² (upper numbers are the reference number of the cited article, not ours).*

With respect to number of tests, we agree with the reviewer that the numbers are relatively low and that not all contacts declared did the test. This may be due to the fact that testing was not obligatory, only quarantine was obligatory. Thus, as in any test registry, there is measurement error and uncertainty on all exposures and outcomes, in this case through lack of test.

The limitation paragraph, from line 294 to 335, emphasize and covers these aspects.

3. I believe that time is an important factor here. E.g., for Delta, the period spans July through November. I would believe the contact patterns are different, as there must be differences in outdoor activities, school holidays, etc. The authors also write that contact tracing effort and test capacity change over time. These temporal changes should be addressed.

Did the definition of contacts change over time or due to vaccination status? Did testing guidelines change over time or due to vaccination status? How many tests were a contact recommended to obtain after being contact traced?

Answer : We agree with the reviewer that the contact patterns changed over time. However, it is not possible to slice the time period too much or we will not have enough data to perform a proper adjusted model to estimate the attack rate.

We added in methods, "Declared contacts" subsection, the information about quarantine obligations and testing associated. Line 385:

Declared contacts in Geneva had the obligation to quarantine during 10 days since the implementation of contact tracing, except for children below 12 years. The 8th February, 2021, it was allowed to shorten the quarantine at day 7 with a negative SarS-CoV-2 PCR test. The quarantine was later shortened to 7 days (31st of December 2021) and to 5 days (12th of January 2022). By end of 2021, vaccinated persons or persons with a positive test during the last 4 month did not have the obligation to quarantine after a contact with an infected index. Since October 2020, health professionals were allowed to work even if quarantined.

Thus, there is measurement error and uncertainty on all exposures and outcomes, in this case through lack of test. We amended the discussion as presented below (line 319):

This propensity of the population to be tested, and the delay between tests and health authority action, varies over time and depends on the health policies implemented. The change in testing is especially visible for children, for whom the testing policies varied from almost no tests during the first waves, even when they were contacts (in part due to recommendations 63 but also because they are often not symptomatic 50) to compulsory autogenic testing in schools if more than two children were infected in a classroom by the end of 2021. The influence of health policies changes can be seen in the increasing delay between the index test result and the first call to contacts with advancing pandemics. Similarly, the introduction of the Swiss sanitary pass the 26th of June 2021 affected the testing of the population. Indeed, since December 2021 it allowed vaccinated or

previously infected patients to use common social venues when non-vaccinated and non-infected persons needed a negative test to do so. Although the adjustment for the propensity to test and for its interaction with the immune status confirmed and even strengthened the effect of immune status on SAR, we cannot completely rule out residual bias, inherent to any observational study.

4. I believe the authors should use the “variant definition period” with >90% dominance cut-off. Using the >50% dominance cut-off allows for too much misclassification across variants. On the other hand, if the author could obtain whole-genome sequence data on the RT-PCR test level, the overlapping periods would allow them to validate their causal model.

Answer : The authors have to disagree on this point. First, the author would like to point out that the variant of concern is not our exposure of interest. We do not intend to prove causality concerning the VoCs, but rather try to adjust for potential confounding (change of viral properties as well as changes of public health policies) by stratifying for the period of VoC dominance. Second, we believe a cutoff at >90% dominance would potentially create a selection bias, by taking out patients in crucial periods. On balance, we believe that the, probably non-differential, misclassification for a stratifying variable is less a problem than the selection bias it would create. Last, the analysis with the >90% cutoff has been added following the reviewer’s comment in the appendix after the last round of review and proved that this did not affect our results.

We amended the discussion as follows line 345:

“Lastly, the study did not assess variant by genotype results based on a PCR test but was based on period of time of variant dominance. Due to an overlap between every variant change, this could alter our results, but probably in a minimal way since variants became dominant quite quickly after they emerged. In addition, though this stratification can be considered a strength since it accounts for differences in vaccination effect across variants, it inflates type I error by using four models without correcting for multiple tests.”

5. The authors define natural immunity as individuals that have a previous positive RT-PCR test. However, there must have been many infected individuals from the first wave that never obtained a positive PCR test. These individuals are then ‘wrongly’ classified as immune-naïve until they become vaccinated or become infected again (identified with a PCR test).

Answer : Again, we agree with the reviewer that there is measurement error and uncertainty on all exposure and outcome. Such misclassification is inherent to any registry-based study. In the present study, the misclassification described by the reviewer would only increase the already important difference observed between immune naïve patients and other types of immunity in our model. We further emphasized this point in the discussion, line 336:

“A third source of information bias is the supposition that the index cases are the primary cases, and that the contacts becoming positive in less than 10 days after their last contact with the index cases are the secondary cases. Though we cannot rule out misclassification (some contacts may actually be the primary cases or tertiary case), the three sensitivity analyses performed to address potential misclassifications indicate that our results are robust to this measurement error. “

6. I am not sure how the authors can argue that the effect of vaccinations is larger on infectiousness compared to susceptibility. The measure for infectiousness is conditional of a breakthrough infection.

Thus, the group of individuals used for the susceptibility estimates is different from the group of individuals used for the infectiousness estimates.

Answer : The authors are not sure to understand the reviewer's point. If the reviewer mean that all index cases are infected while not all contacts are infected, we do agree but do not see the problem. In a study such as our, attack rate is estimated when an index case is infected as the proportion of contacts that are then infected. The effect of index vaccination on the probability of the contact to be infected is the effect of vaccination on infectiousness. The effect of contact vaccination, for the same relation, on the probability of the contact to be infected is the effect of vaccination on susceptibility. In our model, both are thus estimated on the same event for the same dyads and adjusted for several confounders. We thus can compare the effect of vaccination on both infectiousness and susceptibility on the SAR reduction, which is a very common and good measure of viral transmissibility.

Minor comments

1. The paper needs to be proofread.

SARS-CoV-2 is spelled in different ways throughout the paper, e.g., Sars-CoV-2, SARS-Cov-2, SARS-CoV-2, SarS-CoV-2, SARS-CoV2. In the abstract, it is spelled in two different ways!

Line 25, burden should be plural.

Line 220, delta and omicron should be capitalized.

Answer : We corrected the mistake above and we carefully proofread the manuscript.

2. Please provide the estimation equation.

Answer : We added the estimation equation in an appendix to avoid burdening an already long manuscript.

3. I find the age groups to be pretty coarse. How similar are 20 year olds and 60 year olds with respect to access to vaccines, contact patterns (e.g., having small children), health status, etc.?

Answer : We agree with the reviewer that more fine-grained definition of variables would allow a more precise model of reality. The rationale for this age category was to encompass the persons of working age in a main reference category. However:

- *Despite our relatively large sample size, the estimates already have large uncertainty. Adding age category would increase the uncertainty, and we believe without adding much changes.*
- *The access to vaccination is encompassed already in the immunity covariate in our adjustment, the contact pattern with the contact_type variable*

Since age was not our main topic of interest, we decided to tolerate the residual confounding. We clarify this limitation in the discussion (line 341):

"As in any registry study, we cannot rule out residual confounding. Although we did our best to adjust for potential confounders, the categorization we performed on variables such as contact type or age, or the behaviours changes associated with vaccination, could still lead to some confounding. "

4. How are partially vaccinated individuals grouped, e.g., an individuals with first dose but not second dose? Why are booster-vaccinated individuals grouped together with other vaccinated individuals?

Answer : We thank the reviewer for the comment. Indeed, our description was not clear enough. Vaccination is only considered under the influence of the delay since the last dose, irrespective of the

number of doses. Recent literature has shown that the main criteria for the immunity associated with the vaccine is the time since the last dose (see for example <https://www.nature.com/articles/s41467-022-30884-6>, <https://www.nature.com/articles/s41467-022-33096-0> or <https://www.nejm.org/doi/10.1056/NEJMoa2118691>). Despite our relatively large sample size, the estimates already have large uncertainty. To avoid excess of exposure categories and to account for the important time criteria, we choose to not categorize immunity as a function of the doses but as the time since last vaccination dose. The persons were considered as vaccinated if they had at least one dose. More than 95% of vaccinated people had a complete vaccination scheme.

We clarified the immune status operationalization for persons with a single dose in the methods (line 420):

“- Vaccinated more than 6 months or less than 6 months. This category included all persons with at least one dose of a vaccine recognized in Geneva, including booster doses, for which the last date of vaccination was more or less than 6 months.”

and included the information that >95% of people had a complete vaccination scheme.

5. The authors use several different definitions that might or might not refer to the same thing. What is the difference between infectiousness, contagiousness, infectivity, and transmissibility? What is the difference between being contaminated and being infected?

Answer : We used these words as synonyms. We kept infectiousness and infected as it appears to be the more common words.

6. Figure S1. What does “experimental” data mean?

Answer : Experimental was here as opposed to simulated data. We removed experimental.

7. Figure S2 should probably be in fractions of the population.

Answer : We added the fraction of population in a right y axis for the three panels of figure S2

8. Figure S3. Did Geneva not have any school closures?

Answer : Yes, Geneva did, as detailed in supplementary material, in the “Non-pharmaceutical interventions” subsection added in the last round of revision:

- From 16-03-2020 to 11-05-2020: confinement, including schools closure

9. What is the hierarchy of contacts, e.g., is an intimate partner living in the same house classified as “household” or “intimate”?

Answer : He/she would be classified as « household ». We added in methods, line 460: “living under the same roof, having an intimate or familial relationship (but not living under the same roof)”

10. The paper concludes that we need to install ventilation or air filtration and that it is effortless to install. I find no basis for the effect of these measurements nor that it is effortless/free to install in all buildings in Geneva.

Answer : We strongly believe that public health studies should help propose public health policies. As such, we conclude on the effect of vaccination in a “real-world” situation and refer to other literature for the effect of ventilation or air filtration. We provide references for both ventilation and air purification (70-74). The reviewer can also refer to the recommendation of numerous health agencies on the subjects, among which the who <https://www.who.int/news-room/questions-and-answers/item/coronavirus-disease-covid-19-ventilation-and-air-conditioning> or the cdc <https://www.cdc.gov/coronavirus/2019-ncov/community/ventilation.html>.

*Nevertheless, all policies have costs, both in terms of money but also in terms of burden on the society. We of course did not say that ventilation and air filtration does not require effort. We said that such NPIs require no **individual** effort, contrary to other NPIs such as mask wearing or social limitations (our sentence finished with “thus avoiding individual behavioural barriers”). Air purification and proper ventilation evidently require political/institutional efforts, and as any measure they obviously have a cost.*

Because this reviewer has carefully read through the manuscript and still interpreted our sentences in a way we did not intend, we regretfully removed the sentence, since it could clearly be misread.

11. I do not understand this sentence “A coughing index increased the SAR by 4.9pp [4.0, 5.8] and 6.4pp [5.0, 7.9] for the EU1 and alpha variant, but this effect was reduced to 2.3pp [1.2, 3.5] for the delta variant and 1.6pp [0.1, 3.0] for Contact children had a lower SAR, especially for early variant.” (line 167-170)

Answer : Thank you for spotting this typo. The sentence reads:

“A coughing index increased the SAR by 4.9pp [4.0, 5.8] and 6.4pp [5.0, 7.9] for the EU1 and alpha variant, but this effect was reduced to 2.3pp [1.2, 3.5] for the delta variant and 1.6pp [0.1, 3.0] for omicron variant. Contact children had a lower SAR, especially for early variant »

12. Line 178. Reference to Figure S3 should be Figure S6.

Answer :

13. It is very difficult to compare the sensitivity analyses in the appendix to the analyses in the main manuscript.

Answer : We did our best to allow the comparison, by providing the exact same figures and same coefficients tables as in the main text. The authors do not see how they can further ease the comparison.

14. The supplementary appendix is formatted in A3, not in A4.

Answer : Indeed, we choose to have a wide format to have the table or figure in one page, as the format for supplementary material is not constrained.

15. The title of the supplementary appendix uses an apostrophe and not a comma as thousand separator.

Answer : We corrected it

16. I would like more detailed information on the vaccine rollout, including dates (e.g., months) and groups of individuals (e.g., 80+ year, 70+ year). In addition, I failed to find any information of the take-up rate.

Answer : The information about vaccine rollout was already present in the supplementary material, with the exact dates and the group of individuals concerned. The vaccine rollout in Geneva was in beginning only opened for 75+ year old, and then directly for 45+, without intermediary age group. We added the evolution in time of the first dose take-up, as well as the final take-up at the end of the study period. The paragraph in supplementary now reads:

Fig. S3: Evolution in time of the proportion of the population with at least one dose in Geneva

In Geneva, the first vaccination centre opened the 25-01-2021 for person aged 75 or more at first. In February, vaccination was extended to vulnerable persons under 75 (immune-compromised persons, diabetic, ...) and healthcare collaborators. The targeted population was progressively extended to people over 45 years old (from 12-04-2021), people over 16 years old (from 14-05-2021) and finally to children (from 02-07-2021 and from 12-01-2022 for younger people between 5 to 11 years old respectively). The most administrated vaccine type in Geneva was the RNA-based vaccine type, such as Moderna mRNA-1273 (59.83% of the total administrated vaccine doses in Geneva) and Pfizer BNT162b2 (39.86%), other type of vaccine stands for a minor part: Janssen (0.25%) and Nuvaxovid (0.06%). The final uptake at end of February 2022 was of 71.6% of the population who received at least one dose of vaccine (time evolution in Figure S3).

17. Line 92. I do not understand what “two asymptomatic adult men” refers to. Is it both the contact and index case?

Answer : Yes, the reference situation concern the index-contact dyad. We modified the sentence line 92 as follow, line 92:

“The reference category of the contact-index dyad for the adjusted model was defined as follow”

18. I am not sure how to think about nursing homes, jails, etc. I would think the contact patterns and access to tests and isolation is very different in these kind of settings.

Answer : We are not sure what the reviewer means.

If the reviewer means that these setting should be considered separately than other standard situations, we fully agree: that is why we included as covariate the variable “collective structure” in our model (since the first version of our draft) which includes nursing homes and jails. In methods, our article reads:

“The building type were categorised in three categories: building (multi-residential building, potentially having shops), single houses, or collective structure. This last category included nursing homes, jails, asylums and fire-stations.”

If the reviewer means that within the “collective structure” variable, these setting are different one from another, we will answer that we cannot infinitely add new categories to an already complex analysis. The subcategories created would not have enough power to provide any valuable insight.

19. What is the time difference between test and test result?

Answer : Usually a day or less, although this has been time dependent (this time has been larger at the very beginning of the pandemics) and depend on the laboratory too.

20. The type of person that has not been infected or vaccinated is very different over time.

Answer : We do agree with this statement. We added in the limitation the potential residual confounding due to behavioral changes encompassed in immunity: line 341:

“As in any registry study, we cannot rule out residual confounding. Although we did our best to adjust for potential confounders, the categorization we performed on variables such as contact type or age, or the behaviours changes associated with vaccination, could still hinder some confounding.”

21. It seems weird that almost the same proportion of contacts were immune-naïve (NVNI) during Delta and Omicron (53% vs. 54%), cf. Table 1.

Answer : This is actually explainable: Vaccination uptake slowed down during the Delta period to reach ~70% of the population (see figure S3, in response to your point 16). Most of the infections during Delta arrived in the late period, during which the vaccination uptake did almost not change, thus the similar vaccination rate for contact during delta and omicron (this was not the case for index, as vaccination has an important impact on susceptibility)

22. Please briefly state the types of vaccines used in the main manuscript, including the take-up rate. At the moment, the reader has to go to the supplementary appendix.

Answer : We added in the immunity status subsection of the method section, line 429:

“The most administrated vaccine type in Geneva was the RNA-based vaccine type, such as Moderna mRNA-1273 (59.83% of the total administrated vaccine doses in Geneva) and Pfizer BNT162b2 (39.86%), other type of vaccine stands for a minor part: Janssen (0.25%) and Nuvaxovid (0.06%). The final uptake at end of February 2022 was of 71.6% of the population who received at least one dose of vaccine. Details on the vaccination roll-out in Geneva can be found in supplementary materials.